# LEMON 🍋: A Unified and Scalable 3D Multimodal Model for Universal Spatial Understanding

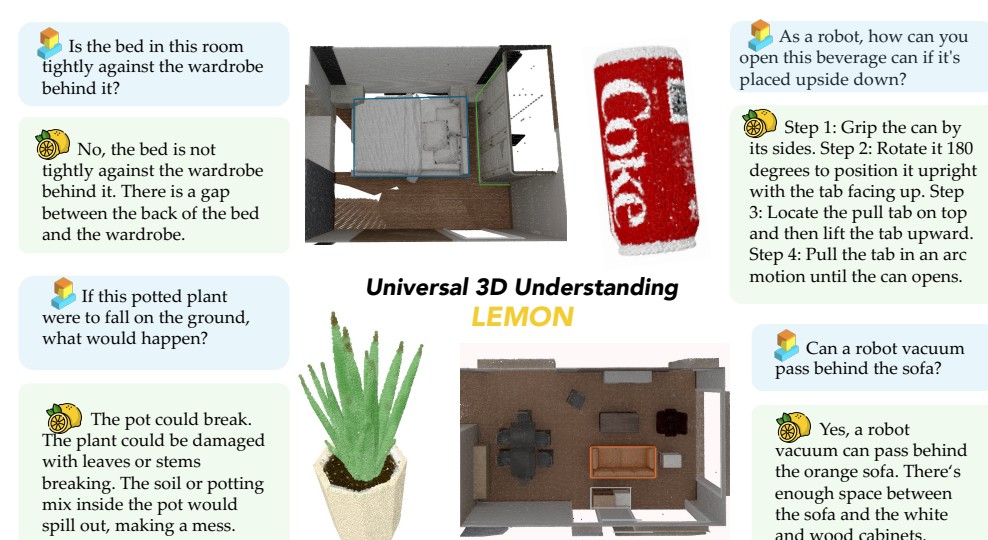

Figure 1: Universal 3D understanding with **Lemon**. **Lemon** demonstrates comprehensive 3D spatial reasoning capabilities across diverse tasks.

## ABSTRACT

Scaling large multimodal models (LMMs) to 3D understanding poses unique challenges: point cloud data is sparse and irregular, existing models rely on fragmented architectures with modality-specific encoders, and training pipelines often suffer from instability and poor scalability. We introduce **Lemon**, a unified transformer architecture that addresses these challenges by jointly processing 3D point cloud patches and language tokens as a single sequence. Unlike prior work that relies on modality-specific encoders and cross-modal alignment modules, this design enables early spatial-linguistic fusion, eliminates redundant encoders, improves parameter efficiency, and supports more effective model scaling. To handle the complexity of 3D data, we develop a structured patchification and tokenization scheme that preserves spatial context, and a three-stage training curriculum that progressively builds capabilities from object-level recognition to scene-level spatial reasoning. **Lemon** establishes new state-of-the-art performance across comprehensive 3D understanding and reasoning tasks, from object recognition and captioning to spatial reasoning in 3D scenes, while demonstrating robust scaling properties as model size and training data increase. Our work provides a unified foundation for advancing 3D spatial intelligence in real-world applications.

## 1 INTRODUCTION

Understanding 3D environments is fundamental for embodied agents, enabling interaction, manipulation, and navigation in the physical world. While large multimodal models (LMMs) have achieved impressive progress in 2D vision-language domains — demonstrated by models such as Flamingo (Alayrac et al., 2022), GPT-4V (OpenAI, 2023) and many open-sourced ones (Chen et al.,

2023; Liu et al., 2024; Zhang et al., 2021; Bai et al., 2025; Peng et al., 2023; Xiong et al., 2024; Yang et al., 2025a; Wang et al., 2025) —scaling such capabilities to 3D data remains an open challenge. The irregular structure, sparsity, and high-dimensional nature of point clouds make 3D learning inherently difficult. Yet, robust 3D understanding is crucial for robotics (Fang et al., 2023; Zhu et al., 2024; Qi et al., 2025), AR/VR systems, and spatial AI (Chen et al., 2024a; Cheng et al., 2024; Zheng et al., 2024a; Yang et al., 2024b; Cao et al., 2024). Despite the emergence of 3D foundation models such as Point-BERT (Yu et al., 2022a) and ULIP (Xue et al., 2022), current efforts fall short of scaling to general-purpose 3D understanding and reasoning tasks in a manner analogous to 2D LMMs.

Most existing 3D LMMs adopt modular designs that employ separate encoders for 3D geometry and language, typically using pretrained 3D encoders such as PointNet++ followed by cross-modal alignment mechanisms (Liu et al., 2023b; Zhou et al., 2023). However, this approach faces several fundamental challenges: (1) 3D encoders are typically pretrained on limited datasets with narrow training objectives, limiting their adaptability to diverse spatial reasoning tasks required by LLMs; (2) unlike the 2D domain where billions of images are available, 3D data remains significantly more constrained in scale, further limiting 3D representation quality; and (3) the architectural imbalance between smaller 3D encoders and large language models creates a representational bottleneck where spatial understanding becomes a performance limitation. Furthermore, reliance on frozen pretrained modality-specific encoders prevents end-to-end optimization and generalization to novel 3D structures, impeding progress toward scalable 3D multimodal learning.

We propose **Lemon**, a unified transformer architecture that directly embeds both 3D geometry and natural language into a shared token space. Rather than relying on separate encoders, **Lemon** treats 3D point cloud patches and language tokens as a unified sequence for joint processing. Each 3D patch is mapped to the language embedding space via a learnable linear projector, and structured using modality-specific and spatial separator tokens. This design allows the model to process spatial and linguistic information cohesively, while eliminating the need for modality-specific encoders and cross-modal alignment mechanisms, improving the scalability of 3D multimodal models. To our knowledge, **Lemon** is the first architecture that unifies point cloud and language processing at the token level within a single transformer for general-purpose 3D reasoning.

To address the challenges of sparse and irregular 3D data, we introduce a dynamic patchification and tokenization strategy. Point clouds are partitioned into patches via a recursive 3D spatial scheme, ensuring uniform patch sizes while preserving geometric structure. Specialized separator tokens encode spatial hierarchy, allowing transformers to operate over structured sequences. To ensure effective learning, we design a three-stage training curriculum: (1) object recognition using large-scale 3D object data extracted from diverse object and scene datasets; (2) object-level captioning and grounding with Cap3D Luo et al. (2023) and GAPartNet (Geng et al., 2023); and (3) scene-level spatial question answering with 3D-GRAND (Yang et al., 2024a). This curriculum supports progressive scaling, transitioning from object-level to complex scene reasoning.

We evaluate **Lemon** across a suite of 3D multimodal tasks, including generative object classification, caption generation, embodied interaction QA, and spatial scene understanding. Our model consistently outperforms prior state-of-the-art baselines in each domain, while exhibiting more favorable scaling behavior as model and data size increase. **Lemon**'s unified architecture reduces parameter redundancy, simplifies the training pipeline, and enables joint spatial-linguistic reasoning, paving the way toward general-purpose 3D multimodal systems for embodied AI, robotics, and beyond.

We summarize our main contributions as follows:

- We propose **Lemon**, the first unified transformer-based 3D LMM that processes point cloud patches and language tokens in a single unified sequence, eliminating the need for modality-specific encoders.
- A dynamic 3D partitioning and tokenization scheme transforms irregular point clouds into structured token sequences, augmented with spatial separator tokens to preserve geometric relationships.
- Our three-stage progressive training curriculum enables stable and scalable 3D LMM learning, advancing from object recognition to captioning and finally to scene-level spatial reasoning with stage-specific optimization strategies.
- Extensive experiments across diverse 3D understanding and reasoning tasks demonstrate consistent improvements over existing 3D LMMs and favorable scaling behavior with model size and data.

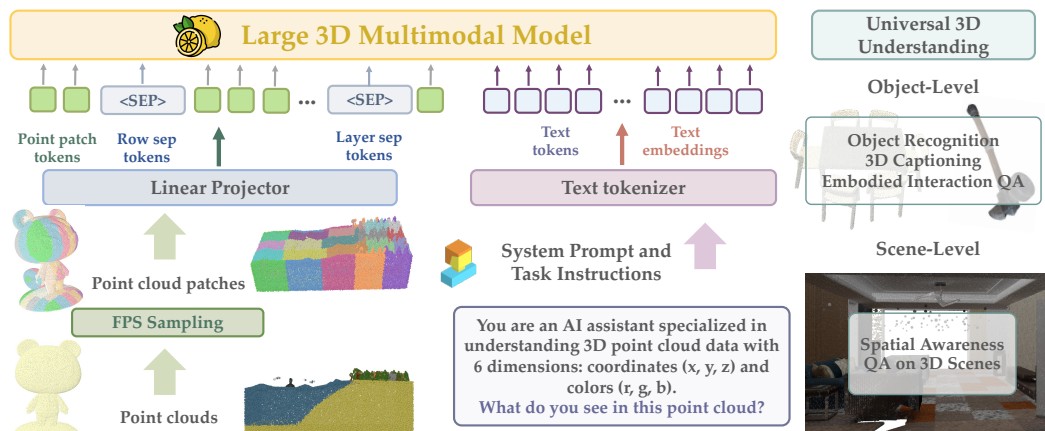

Figure 2: Method overview. **Lemon** processes point clouds using FPS sampling and dynamic patchification, feeding point patch tokens (representing projected 3D patch embeddings) and text tokens into a unified Large 3D Multimodal Model to handle both object-level (e.g., recognition, captioning and embodied interaction QA) and scene-level (e.g., spatial QA) tasks. Unlike existing methods, **Lemon** leverages a single framework to enhance cross-modal alignment and multi-scale adaptability.

## 2 LEMON: LEARNING A UNIFIED AND SCALABLE 3D MULTIMODAL MODEL

We present **Lemon**, which integrates the 3D modality and language in a unified transformer to process point cloud patches and language tokens. **Lemon** enables more effective scaling laws, allowing 3D representation capabilities to grow naturally with increasing training data. To achieve stable training for this unified architecture, we design a comprehensive training pipeline with carefully orchestrated strategies for progressive scaling and balanced multi-modality training.

### 2.1 MODEL ARCHITECTURE

As illustrated in Figure 2, **Lemon** employs a unified transformer architecture that fundamentally differs from traditional multimodal models by directly processing 3D spatial information within the language model framework. Rather than utilizing separate 3D encoders followed by cross-modal alignment modules, **Lemon** integrates point cloud patch processing and language understanding in a single transformer.

The architecture processes point cloud patches through a learnable linear projector that maps each patch to continuous embeddings compatible with the language model's embedding space. We introduce specialized tokens for 3D modality encoding: `<pointcloud>` and `</pointcloud>` mark the boundaries of point cloud sequences, while `<point_patch>` denotes individual point cloud patches. Additional separator tokens `<layer_sep>`, `<row_sep>` are employed to maintain spatial structure within the point cloud sequences.

The integration strategy concatenates 3D patch embeddings with text token embeddings, creating a unified sequence that flows through a single transformer. This design facilitates seamless integration of spatial and linguistic information, allowing unified processing of both modalities within a shared representational space. This unified design simplifies the overall architecture by eliminating separate modality encoders commonly used in heterogeneous approaches.

**Hierarchical Spatial Partitioning.** Our patchification process operates through a hierarchical three-dimensional partitioning scheme that divides point clouds along Z→Y→X axes in sequence. Given a point cloud $\mathcal{P} = \{p_i \in \mathbb{R}^3\}_{i=1}^N$, we define $M$ as the target number of points per patch and $K$ as the maximum number of splits per axis.

The number of splits for each axis is determined adaptively based on point distribution:

$$\text{splits}_{\text{axis}} = \min\left(K, \left\lfloor \frac{N_{\text{total}}}{N_{\text{target}}} \right\rfloor\right) \tag{1}$$

where $N_{\text{total}}$ is the total number of points and $N_{\text{target}}$ decreases hierarchically: $N_{\text{target}} = M \times K^2$ for Z-layers, $N_{\text{target}} = M \times K$ for Y-rows, and $N_{\text{target}} = M$ for X-patches.

Once the number of splits is determined for each axis, we divide the coordinate range into equal intervals to create spatial regions:

$$\mathcal{P}^{(z_k, y_j, x_l)} = \{p_i \in \mathcal{P} | z_k \leq p_i^z < z_{k+1}, y_j \leq p_i^y < y_{j+1}, x_l \leq p_i^x < x_{l+1}\}, \tag{2}$$

where $(z_k, y_j, x_l)$ represents the coordinate indices with $k \in [0, \text{splits}_z)$, $j \in [0, \text{splits}_y)$, and $l \in [0, \text{splits}_x)$, and the boundary values are computed by equally dividing each axis range by the corresponding number of splits.

**Patch Standardization.** We enforce uniform patch size $|\mathcal{P}^{(z_k, y_j, x_l)}| = M$ through strategic point replication for insufficient patches and Farthest Point Sampling (FPS) for oversized patches. FPS iteratively selects the next point $p_{\text{next}}$ that maximizes the minimum distance to all previously selected points:

$$p_{\text{next}} = \arg \max_{p \in \mathcal{P}^{(z_k, y_j, x_l)} \setminus \mathcal{S}} \min_{q \in \mathcal{S}} \|p - q\|_2 \tag{3}$$

where $\mathcal{S}$ denotes the set of already selected points.

**Spatial Token Organization.** To preserve 3D spatial relationships, patches are sorted by $(z, y, x)$ coordinates with separator tokens: <layer_sep> for Z-coordinate changes, <row_sep> for Y-coordinate changes within layers, and <point_patch> for individual patch positions. A concrete example is provided in Appendix D.1.

Based on empirical analysis of typical point cloud datasets and compatibility requirements, we set $M = 512$ and $K = 5$. These parameters accommodate the majority of point cloud data distributions while ensuring patch embeddings align with standard transformer dimensions ($M \times 6 = 3072$ dimensions, compatible with 2D VLM architectures). We validate these choices through comprehensive ablation studies in our experiments.

## 2.2 TRAINING PARADIGM

We present a three-stage training curriculum designed to progressively develop 3D spatial understanding capabilities while maintaining language comprehension.

**Stage 1: Object Recognition.** The initial stage focuses on establishing fundamental 3D object recognition capabilities through large-scale classification tasks. We train **Lemon** to predict object category labels conditioned on 3D patches, enabling the model to learn the semantic meaning of our specialized tokens. This stage utilizes diverse 3D object datasets, including Objaverse (Deitke et al., 2023) and objects extracted from various synthetic and real-world scene datasets, providing comprehensive exposure to geometric variations and object categories. Similar to the alignment training for 2D LMMs, this stage proves crucial for developing meaningful 3D representations that are aligned with language models and serve as the foundation for subsequent training phases.

Table 1: Dataset statistics for each training stage.

**Stage 2: Object Captioning and Grounding.** Building upon the recognition capabilities established in Stage 1, we transition to object-level caption generation tasks. This stage teaches the model to articulate spatial properties and geometric characteristics of individual 3D objects in natural language. The training data consists of high-quality caption annotations from Cap3D (Luo et al., 2023) and detailed object grounding data from GAPartNet (Geng et al., 2023), en-

| Stage | Dataset Sources | Language Pairs | Point Clouds |
|-------|-----------------|----------------|--------------|
| Stage 1 | Objaverse, ProcThor, ScanNet, ShapeNet, MultiScan, Structured3D, 3RScan, ARKitScenes, HM3D, 3D-FUTURE | 1.87M | 1.87M |
| Stage 2 | Cap3D-ShapeNet, Cap3D-Objaverse, Cap3D-ABO, GAPartNet | 140K | 140K |
| Stage 3 | 3D-GRAND: Scene Spatial QA datasets, 30% of Stage 2 | 142K | 50K |

abling the model to bridge the gap between geometric understanding and language generation. This intermediate phase prepares the model for more complex spatial reasoning tasks while preserving the object-level understanding acquired previously.

**Stage 3: Scene Spatial Question Answering.** The final stage elevates the model's capabilities from object-level understanding to comprehensive scene-level spatial reasoning. We introduce complex

question-answering tasks that require understanding spatial relationships, object interactions, and scene-level context from 3D-GRAND (Yang et al., 2024a). The training data encompasses diverse question types, from basic object localization to complex spatial relationships and scene interpretation. To preserve object-level capabilities, we also incorporated a portion of object-level instruction data into the Stage 3 training mixture. This stage culminates in instruction tuning that enables versatile 3D understanding across various spatial reasoning tasks, from object-level queries to sophisticated scene analysis.

Our training curriculum is grounded in two fundamental design principles. First, we implement a progressive learning paradigm that transitions from object-level to scene-level understanding, ensuring the model first masters individual geometric structures before tackling intricate spatial relationships. Second, we employ a complexity-driven approach that advances from basic recognition tasks to spatial reasoning capabilities, enabling the model to develop universal 3D understanding through systematic skill acquisition.

**Training infrastructure.** We implement **Lemon** using LLaMA-Factory (Zheng et al., 2024b) as our training framework, with modifications to support 3D point cloud patch inputs and our specialized tokenization scheme. All experiments are conducted on an 8×H100 cluster. We adopt standard learning rate schedules with cosine decay and appropriate warm-up strategies for each training stage. Detailed training hyperparameters and efficiency analysis are provided in Appendix C and D.8. To foster research in 3D multimodal learning, we will release all training datasets, code, and model weights as open-source resources.

## 3 EXPERIMENTS

Our extensive experiments evaluate **Lemon** across three key dimensions of 3D understanding: embodied object interaction, scene-level spatial reasoning, and fundamental 3D object recognition and captioning. These evaluations demonstrate **Lemon**'s spatial intelligence capabilities as a generalist 3D multimodal model.

### 3.1 SETUP

**Model Implementations.** We implement **Lemon** based on Qwen2.5-7B-Instruct (Bai et al., 2025). The model maintains the original language modeling capabilities while extending to process 3D spatial inputs through our specialized tokenization scheme and architecture modification.

**3D LMMs Baselines.** We compare against several state-of-the-art 3D language multimodal models, including object-focused LLMs: PointLLM (Xu et al., 2024) and ShapeLLM (Qi et al., 2024), and scene-oriented LLMs: 3D-LLM (Hong et al., 2023), Ll3da (Chen et al., 2024b), LEO (Huang et al., 2023), and LSceneLLM (Zhi et al., 2024). Since object-focused models (PointLLM/ShapeLLM) utilize 3D object datasets with substantial overlap to ours but with different training pipelines, we fine-tune them for 2 epochs using our scene spatial QA datasets to ensure fair comparison across all spatial reasoning tasks.

**2D VLM Baselines.** To assess the advantages of native 3D processing, we evaluate on strong 2D VLMs: LLaVA-1.5-13B (Liu et al., 2024), and Qwen2.5-VL-7B (Bai et al., 2025), GPT-4V(vision) (OpenAI, 2023). For these models, we provide random single-view or multi-view rendered images generated from the point cloud datasets as input. All open-source models undergo fine-tuning for 2 epochs on the rendered 2D data to optimize their performance on our benchmarks.

**Benchmarks.** To systematically evaluate our model's 3D understanding capabilities, we employ a multi-level evaluation strategy. For advanced embodied interaction understanding, we conduct zero-shot evaluation on 3D MM-Vet (Qi et al., 2024), which encompasses object-level embodied task planning and decomposition. At the scene level, we emphasize spatial awareness evaluation in 3D spaces, requiring models to understand spatial relationships between scene objects rather than object recognition in scenes. We evaluate on the established 3D-GRAND (Yang et al., 2024a) binary and non-binary spatial QA test sets, which offers greater scene diversity than previous scene benchmarks (Ma et al., 2022; Azuma et al., 2022), and further incorporate 100 challenging embodied 3D spatial QA questions (detailed in Appendix E.1, including distance estimation (e.g., comparing distances between multiple objects), navigability analysis (e.g., determining if a robot can pass through a gap), and collision avoidance, for zero-shot evaluation of spatial reasoning.

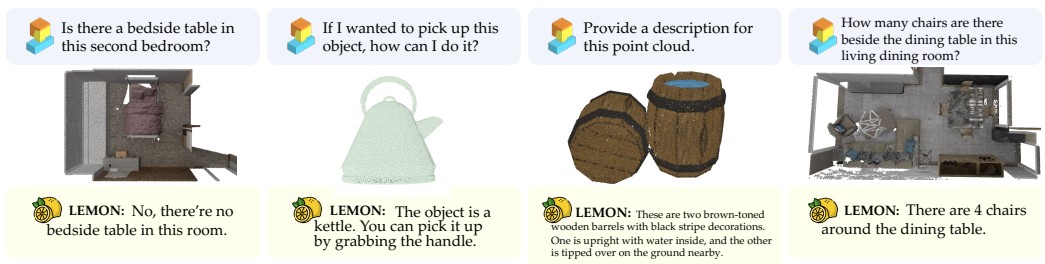

In this children's room, is the flower chair closer to the bed or to the pink desk?

**GPT-4V:** Based on the image, the flower chair is closer to the bed. It appears to be placed almost directly at the foot of the bed, while the pink desk is in front of it, with a noticeable gap between them.

**LEMON:** The flower chair is flush against the pink desk. Therefore, it is closer to the pink desk than to the bed.

If I am sitting on the long sofa, can I pass through the gap between the coffee table and the short sofa to reach the dinner desk?

**GPT-4V:** It seems unlikely. this gap is very narrow in the figure. If you wanted to get from the sofa to the dining table/bar, you would need to get up and walk around the coffee table.

**LEMON:** Yes, you can pass between the sofa and the coffee table.

Figure 3: Comparison of **Lemon** and GPT-4V on 3D scene understanding. **Lemon** leverages 3D point clouds to accurately capture spatial relationships, such as the flower chair's position and sofa passability, outperforming GPT-4V, which struggles with 2D images by missing depth and occlusion details. This highlights **Lemon**'s superior 3D scene comprehension.

Is there a bedside table in this second bedroom?

If I wanted to pick up this object, how can I do it?

Provide a description for this point cloud.

How many chairs are there beside the dining table in this living dining room?

**LEMON:** No, there're no bedside table in this room.

**LEMON:** The object is a kettle. You can pick it up by grabbing the handle.

**LEMON:** These are two brown-toned wooden barrels with black stripe decorations. One is upright with water inside, and the other is tipped over on the ground nearby.

**LEMON:** There are 4 chairs around the dining table.

Figure 4: Examples of **Lemon** in diverse 3D understanding tasks. **Lemon** demonstrates its capability by accurately addressing object-level tasks(e.g., object description and interaction guidance) and scene-level tasks (e.g., spatial analysis of room elements). Its unified framework ensures versatility across various 3D understanding tasks.

For fundamental capabilities assessment, we evaluate object recognition and detailed captioning performance. Beyond using the widely adopted benchmark Objaverse-LVIS (Deitke et al., 2023) for object-level evaluation, we include 2,000 unseen objects extracted from 5 various scene datasets (detailed in Appendix E.2) to ensure comprehensive evaluation across diverse object categories and provide more representative results.

**Evaluation Protocol.** To ensure reproducibility and facilitate fair comparison, we categorize our evaluation protocols into traditional, learning-based, and LLM-as-judge metrics. For the object recognition tasks reported in Table 3, we utilize an LLM-assisted accuracy metric. Instead of strict string matching, we employ GPT-4 (Achiam et al., 2023) to determine semantic correctness by verifying if the predicted class name is semantically equivalent to the ground truth label. For object captioning, we report learning-based metrics, specifically Sentence-BERT (Reimers & Gurevych, 2019) and SimCSE (Gao et al., 2021), which compute the cosine similarity between the embeddings of the generated and reference captions. We complement this with an LLM-as-judge evaluation, where GPT-4 scores the comprehensive quality of the captions. For the scene-level tasks in Table 2, we employ binary accuracy for discriminative questions and GPT-4 scores for open-ended generation. All specific prompts used for these evaluations are detailed in the Appendix F.

## 3.2 MAIN RESULTS

**Embodied Interaction Comprehension on 3D Objects.** We evaluate **Lemon**'s performance on 3D MM-Vet using GPT-4 (Achiam et al., 2023) as the evaluator, which assesses core 3D understanding capabilities including visual recognition, knowledge reasoning, language generation, spatial awareness, and embodied interaction. As shown in Table 2, **Lemon**-7B achieves the highest performance among all 3D language multimodal models, significantly outperforming existing strong 3D LMM baselines such as ShapeLLM-13B and PointLLM-13B while using only 7.63B parameters, demonstrating superior parameter efficiency. The performance gap between **Lemon** and the strongest 2D

| Model | Trainable Params | Input | Embodied Object QA | Scene Spatial Awareness QA | |
|---|---|---|---|---|---|
| | | | GPT4 | Binary Accuracy | GPT4 |
| LLaVA-1.5-13B | 13.03B | Single-view Img. | 47.3 | 57.62 | 40.18 |
| LLaVA-1.5-13B | 13.03B | Multi-view Img. | 50.7 | 59.8 | 41.2 |
| Qwen2.5-VL-7B | 7.61B | Single-view Img. | 52.23 | 64.32 | 47.56 |
| Qwen2.5-VL-7B | 7.61B | Multi-view Img. | 55.9 | 69.1 | 49.3 |
| GPT-4V | - | Single-view Img. | 57.41 | 69.23 | 52.34 |
| GPT-4V | - | Bird-view Img. | 58.21 | 71.18 | 53.72 |
| GPT-4V | - | Multi-view Img. | 63.40 | 75.32 | 53.68 |
| PointLLM-7B | 7.01B | 3D Point Cloud | 41.20 | - | - |
| PointLLM-13B | 13.01B | 3D Point Cloud | 46.63 | - | - |
| ShapeLLM-7B | 7.04B | 3D Point Cloud | 47.42 | 58.49 | 41.39 |
| ShapeLLM-13B | 13.04B | 3D Point Cloud | 53.15 | 60.27 | 42.34 |
| 3D-LLM | - | 3D Point Cloud | 38.36 | 51.25 | 33.43 |
| Ll3da | 1.3B | 3D Point Cloud | - | 53.45 | 39.60 |
| LEO | 7.01B | 3D Point Cloud | 39.28 | 49.74 | 30.29 |
| LSceneLLM | | 3D Point Cloud | 38.54 | 65.46 | 45.79 |
| **Lemon-7B** | 7.63B | 3D Point Cloud | **57.22** | **74.32** | **53.45** |

Table 2: Performance comparison on Embodied Object QA and Scene Spatial Awareness QA benchmarks across 2D vision-language models and 3D multimodal models.

vision-language model (VLM) GPT-4V is minimal, demonstrating **Lemon**'s solid understanding of both the intrinsic properties and practical applications of 3D objects.

**Spatial Awareness in 3D Scenes.**   For scene-level spatial reasoning, we evaluate **Lemon** on 3D-GRAND benchmarks that focus on understanding spatial relationships between objects within 3D environments. As demonstrated in Table 2, **Lemon** achieves exceptional performance in both binary accuracy and GPT-4 evaluation on non-binary QA, substantially outperforming all existing 3D multimodal models.  Our unified architecture enables **Lemon** to excel in spatial reasoning tasks, achieving notable gains of 8.9% and 7.7% in binary accuracy and non-binary QA performance respectively over the next best 3D baseline model.  This demonstrates that our model not only understands spatial structures but also maintains superior language generation capabilities, enabling precise spatial reasoning outputs.

Importantly, **Lemon** surpasses GPT-4V with random single-view images as inputs.  s illustrated in Figure 3, **Lemon** leverages 3D point clouds to accurately capture spatial relationships, such as furniture positioning and navigational possibilities, whereas GPT-4V struggles with 2D images due to missing depth and occlusion details. Figure 4 further demonstrates **Lemon**'s versatility across diverse 3D understanding tasks, from object-level reasoning to complex scene analysis, significantly reducing spatial hallucinations commonly observed in 2D LLMs when processing 3D environments. This highlights how 3D inputs provide complete geometric information without viewpoint limitations that inherently constrain 2D representations. Our method achieves comparable performance with the closed-source model using multi-view inputs while outperforming all open-source models, which fully demonstrates the critical importance of open-sourced 3D LMMs for advancing spatial intelligence capabilities. As an open-source model, **Lemon** demonstrates substantial potential for further scaling with the emergence of larger and more diverse 3D datasets, paving the way for even more capable 3D LMMs that can unlock the unlimited potential of 3D spatial reasoning in real-world applications.

**3D Object Generative Recognition and Captioning.**   As shown in Table 3, **Lemon** demonstrates strong performance across both tasks. In object recognition, **Lemon** achieves results comparable to the best 2D VLM GPT-4V while significantly outperforming all 3D LMMs. For object captioning, **Lemon** substantially exceeds other 3D multimodal models across all metrics, showcasing its ability to generate detailed and accurate textual descriptions. These results validate **Lemon**'s robust understanding of 3D object properties, establishing it as a capable foundation for both spatial recognition and linguistic articulation while achieving performance on par with leading 2D vision-language models.

Additional experiments in the Appendix D.6 demonstrate our model's robustness on sparse or noisy point clouds, superior performance on 3D visual grounding tasks, and consistent advantages in zero-shot evaluations on ScanQA (Azuma et al., 2022) and SQA3D (Ma et al., 2022) benchmarks.

| Model | Trainable Params | Input | Recognition Accuracy | Object Captioning | | |
| --- | --- | --- | --- | --- | --- | --- |
| | | | | Sentence-BERT | SimCSE | GPT-4 |
| LLaVA-1.5-13B | 13.03B | Single-view Img. | 36.04 | 38.89 | 40.54 | 17.20 |
| Qwen2.5-VL-7B | 7.61B | Single-view Img. | 58.72 | 52.74 | 54.33 | 52.04 |
| GPT-4V | - | Single-view Img. | 61.33 | 57.63 | 58.72 | 56.89 |
| 3D-LLM | - | Multi-view Img. | 22.89 | 42.13 | 42.79 | 32.60 |
| PointLLM-7B | 7.01B | 3D Point Cloud | 53.49 | 47.33 | 47.93 | 40.78 |
| PointLLM-13B | 13.01B | 3D Point Cloud | 54.32 | 47.67 | 48.22 | 40.39 |
| ShapeLLM-7B | 7.04B | 3D Point Cloud | 54.09 | 47.63 | 49.35 | 45.81 |
| ShapeLLM-13B | 13.04B | 3D Point Cloud | 54.15 | 47.80 | 49.21 | 46.09 |
| MiniGPT-3D | 2.7B | 3D Point Cloud | 53.52 | 47.64 | 47.20 | 45.78 |
| GreenPLM | 3.8B | 3D Point Cloud | 54.65 | 48.72 | 48.40 | 42.78 |
| **Lemon-7B** | 7.63B | 3D Point Cloud | **59.20** | **52.23** | **53.59** | **50.76** |

Table 3: Evaluation results on fundamental 3D Object Recognition and Captioning tasks across 2D vision-language models and 3D multimodal models.

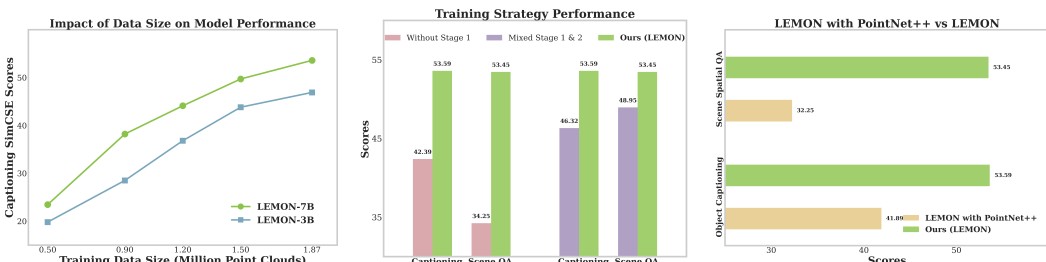

(a) Scaling laws for 3D LMMs across different training data sizes.  (b) Model performance comparison of different training strategies.  (c) Comparing **Lemon** with and without PointNet++ encoder.

Figure 5: Ablation studies on key design choices in **Lemon**.

## 3.3 ABLATION STUDIES

**Scaling Laws in 3D LMMs.** To our knowledge, this work presents the first systematic analysis of scaling laws in 3D multimodal language models. We train an additional 3B model based on Qwen2.5-3B-Instruct, conducting only the first two training stages due to resource constraints, allowing us to evaluate scaling behavior through captioning performance. Figure 5a demonstrates how **Lemon**'s performance scales with training data size from our Stage 1 pretraining, using captioning performance as a representative metric to evaluate scaling behavior. The analysis reveals clear power-law scaling behavior for both **Lemon-7B** and **Lemon-3B**, with consistent performance improvement across 0.5 million to 1.87 million point cloud samples. **Lemon**'s unified design enables straightforward scaling analysis, avoiding the complexity of heterogeneous architectures that require additional parameter allocation laws.

Our analysis is based on Stage 1 object classification data, and we anticipate that introducing more diverse and richer 3D-language paired datasets could achieve further performance gains beyond what is shown in this scaling study. Our findings suggest that coordinated scaling of model size and training data follows predictable patterns in 3D multimodal learning, providing insights that may inform more efficient resource allocation in future 3D LMM development.

**Isolating Architectural Benefits.** To strictly evaluate architectural contributions, we retrained **Lemon** using the same Vicuna-7B-1.1 (Touvron et al., 2023) with LLaMA backbone and training data as ShapeLLM-7B (Qi et al., 2024) As shown in Table 4, **Lemon** consistently outperforms ShapeLLM under these identical settings ("All Same"), achieving gains of +2.4 in object captioning, +5.9 in 3D MM-Vet, and +5.4 in Scene Spatial QA. These results confirm that our unified transformer architecture is the primary driver of performance by eliminating the bottleneck imposed by separate 3D encoders. Furthermore, the consistent performance growth observed when utilizing our training data demonstrates the robust data scaling capabilities of **Lemon**. Finally, the superior results obtained with the Qwen2.5-7B (Bai et al., 2025) backbone indicate that stronger language models also significantly contribute to enhancing 3D multimodal performance.

**Lemon benefits from the training curriculum.** To validate our three-stage training curriculum, we conduct ablation studies comparing different training strategies on object captioning and scene

Table 4: **Controlled comparison isolating architectural benefits.** We align the LLM backbone and training data with ShapeLLM to strictly evaluate the contribution of our unified architecture. "All Same" denotes retraining **Lemon** using the exact same backbone and data source as the baseline.

| Model | LLM Backbone | Training Data | Obj. Cap. (SimCSE) | Embodied QA (3D MM-Vet) | Scene Spatial QA (GPT-4) |
|---|---|---|---|---|---|
| ShapeLLM-7B | Vicuna-1.1 7B | ShapeLLM Data | 49.4 | 47.4 | 41.39 |
| **Lemon-7B (All Same)** | Vicuna-1.1 7B | ShapeLLM Data | 51.8 | 53.3 | 46.80 |
| **Lemon-7B (Same Arch.)** | Vicuna-1.1 7B | **Lemon** Data | 52.4 | 53.2 | 50.60 |
| **Lemon-7B (Default)** | Qwen2.5-7B | **Lemon** Data | **53.6** | **57.2** | **53.45** |

QA tasks. Figure 5b demonstrates significant impact of our progressive training approach, evaluated using SimCSE and GPT-4 metrics. We compare three variants: training without Stage 1, mixed Stage 1 & 2 training, and our complete three-stage curriculum. The results reveal substantial performance gaps across both tasks. Without Stage 1 initialization, the model underperforms compared to our complete curriculum, while the mixed approach shows improvement but still falls short, suggesting that progressive learning is more effective than joint training of different task types. Our analysis demonstrates that Stage 1 with large-scale 3D data serves as crucial foundation, enabling the model to learn fundamental 3D spatial representations and specialized token semantics. This progressive curriculum allows **Lemon** to develop robust 3D understanding capabilities in a structured manner.

**3D encoder is not necessary in 3D LMMs.** To investigate the necessity of dedicated 3D encoders, we conduct a controlled experiment using only xyz coordinates without RGB information. Following previous practices where PointNet++ is commonly used as the 3D encoder in existing 3D LMMs, we modify our architecture to process point cloud patches through PointNet++(Qi et al., 2017b) before our linear projector, freezing PointNet++ parameters while keeping other training configurations identical. Figure 5c reveals that adding PointNet++ actually degrades performance across both tasks, challenging the assumption that specialized 3D encoders are necessary for effective 3D-language understanding. We attribute this performance degradation to two key factors: (1) PointNet++ is pretrained on limited 3D object datasets for shape classification, lacking semantic alignment with language-oriented tasks, and (2) the frozen encoder introduces a representational bottleneck that prevents end-to-end optimization. Our unified architecture demonstrates that direct processing through the language model framework can effectively learn task-relevant 3D representations, supporting our design choice of eliminating separate 3D encoders in favor of fully integrated joint optimization. Additional ablation studies are provided in the Appendix D.

## 4 RELATED WORKS

**Multimodal Large Language Models**   Building upon the advances of recent large language models (LLMs) (Touvron et al., 2023; Zhang et al., 2023; Brown et al., 2020; Bai et al., 2023), numerous works (Chen et al., 2023; Liu et al., 2024; Li et al., 2022; Liu et al., 2023a; Wu et al., 2023; Gong et al., 2023; Driess et al., 2023; Wang et al., 2024b;a; Yang et al., 2025a)have investigated multimodal large language models (MLLMs) capable of understanding both visual and textual inputs. Although MLLMs excel at numerous 2D vision-language tasks, their ability to understand complex 3D world is still an open question. In the 2D domain, unified architectures like VisualBERT (Li et al., 2019), Fuyu-8B (Li et al., 2023) and SOLO (Chen et al., 2024c) have demonstrated the potential of processing image patches and language tokens within a single Transformer. However, extending such unified approaches to 3D presents additional challenges due to the irregular structure of point clouds and the limited availability of 3D-language paired data. Existing 3D MLLMs can be broadly categorized into two paradigms. One line of work (Guo et al., 2023; Yang et al., 2025b; Qi et al., 2024) directly encodes raw 3D data. However, this late-stage alignment approach struggles to capture the intricate relationship between 3D data and language. Moreover, the scarcity of 3D data limits the encoder (Qi et al., 2017a;b; 2023)'s representational capacity and generalization ability, leading to suboptimal performance, particularly in complex scenarios. **Lemon** overcomes these limitations through a unified Transformer architecture, early fusion, and dynamic point cloud patchification, significantly enhancing cross-modal alignment capability, 3D representation capability, and model scalability.

**3D Understanding with LLM**   The challenges of 3D understanding lie in identifying the semantic meanings, physical properties and spatial relationship of objects. Existing works Qi et al. (2024);

Guo et al. (2023); Hong et al. (2023); Chen et al. (2024b); Yu et al. (2022b) explored leveraging the remarkable perceptual and reasoning capabilities of LLMs to enhance the understanding of 3D point clouds. Existing models typically focus on a single scale. For object-/part-level 3D understanding related works such as PointLLM Xu et al. (2024), ShapeLLM Qi et al. (2024) and MiniGPT-3D (Tang et al., 2024) can identify the semantic and physical properties of individual objects, such as shape and material. However, when faced with scene-level point clouds that include multiple objects and complex spatial relationships, these models often struggle to capture the interactions between objects and the overall context, leading to a decline in performance. For scene-level understanding, previous works Zhi et al. (2024); Yang et al. (2025b); Azuma et al. (2022); Jiao et al. (2022); Ma et al. (2022); Parelli et al. (2023); Chen et al. (2024d) excel at understanding multiple objects and their spatial relationships, capable of handling the overall layout of scenes. Some recent approaches explore alternative representations: Video-3D LLM Zheng et al. (2025) treats 3D scenes as 2D video projections to leverage video-LLMs, while Inst3D-LMM Yu et al. (2025) relies on a complex multi-stage pipeline utilizing external 3D instance segmentors. However, these projection-based or multi-stage methods often suffer from geometric information loss or error propagation from external modules. Moreover, scene-oriented models typically rely on large amounts of annotated scene data, limiting their generalization ability and making it difficult to adapt to diverse application scenarios. **Lemon**, through unified design, overcomes the aforementioned limitations and possesses significant multi-scale adaptability, enabling it to efficiently handle both individual object point clouds and scene-level point clouds.

## 5 CONCLUSION AND DISCUSSIONS

In this paper, we introduce **Lemon**, a unified transformer architecture that successfully addresses the challenge of scaling multimodal learning to 3D spatial understanding. By processing point cloud patches and language tokens within a single sequence, **Lemon** eliminates the complexity of heterogeneous architectures and achieves state-of-the-art performance across diverse 3D multimodal tasks, from object recognition to complex spatial reasoning. Our comprehensive experiments demonstrate that **Lemon** not only outperforms existing 3D LMMs but also exhibits favorable scaling behaviors, providing the first systematic analysis of scaling laws in 3D multimodal learning. Future directions include developing fine-grained 3D grounding capabilities, exploring cross-modal alignment techniques, and integrating with embodied AI agents for real-world robotics applications. We believe **Lemon**'s unified approach opens new possibilities for scalable 3D multimodal learning, providing a solid foundation for future research in spatial intelligence and embodied AI.

## REPRODUCIBILITY STATEMENT

We are committed to ensuring the reproducibility of our results. Detailed descriptions of our experimental setup, including model architectures, training procedures, and hyperparameter settings, are provided in Appendix C.

## LLM USAGE STATEMENT

We confirm that Large Language Models (LLMs) were exclusively utilized for minor editing, polishing, and improving the clarity and flow of the text within this paper. LLMs were not employed for generating any core content, scientific ideas, or experimental results. All original contributions, including concepts, methodologies, and findings, are solely the work of the authors.

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

Table 5: Implementation Details for **Lemon** Training

| Hyper-parameter | Value |
| --- | --- |
| base model | Qwen/Qwen2.5-7B-Instruct |
| batch size | 512 |
| learning rate | 1.0e-5 |
| num train epochs | 3 |
| lr scheduler type | cosine |
| warmup ratio | 0.1 |
| bf16 | true |

## A  LIMITATIONS AND BROADER IMPACT

**Limitations.**  The investigation into large-scale 3D multimodal modeling using a unified transformer architecture remains nascent. Current limitations include substantial computational requirements for training and inference, and dependency on limited 3D-language paired datasets compared to 2D counterparts. The point cloud patch tokenization may also introduce discretization artifacts that affect fine-grained spatial reasoning, and the model's performance can be sensitive to point cloud quality and density variations. Continued advancements in unified 3D multimodal architectures, more efficient training strategies, and larger-scale 3D datasets are anticipated to address these challenges.

**Broader Impact.**  Although developing 3D multimodal models with strong capabilities brings significant advancements in spatial AI and robotics applications, enabling more natural human-robot interaction and enhanced accessibility tools, it also poses potential negative impacts. One concern is the risk of misuse, where the model could be employed for malicious purposes, such as generating misleading 3D content or facilitating unauthorized surveillance in physical environments. Additionally, the model may inadvertently exacerbate biases present in the 3D training data, leading to unfair or discriminatory outcomes in spatial reasoning tasks and embodied AI applications.

## B  EXTENDED BACKGROUND

**3D modality and language alignment.**  Large language models (LLMs) have been extensively employed in various works for 3D shape and space understanding, leveraging point clouds (Qi et al., 2024; Chen et al., 2024b; Hong et al., 2023; Zhu et al., 2023), RGBD images (Cheng et al., 2024), and other 3D representations (Yang et al., 2024a) as input. These 3D modalities provide crucial geometric and structural information that enables more comprehensive scene understanding and object manipulation in complex environments. These approaches aim to endow models with the capability to comprehend 3D data and perform spatial reasoning, thereby addressing tasks that cannot be effectively solved using 2D images alone. Similar to vision-language models, a fundamental challenge in building effective 3D-language models is establishing robust cross-modality alignment between 3D features and language features. This alignment is critical as it directly impacts the model's ability to connect language descriptions with corresponding 3D structures, determining performance across 3D understanding tasks.

**Scalability challenges in 3D LMMs.**  Current approaches to 3D-language alignment typically employ pretrained 3D encoders, such as PointNet (Qi et al., 2017a), PointNet++(Qi et al., 2017b), or develop specialized encoders through contrastive learning paradigms as demonstrated in ReCon++ (Qi et al., 2024) and point embeddings(Chen et al., 2024d). These 3D encoders still exhibit significant limitations in adapting to novel 3D data distributions and more complex spatial reasoning tasks, primarily because they are trained on narrow data distributions with restricted training objectives. Unlike the 2D domain where billions of images are available for training Radford et al. (2021), the 3D data landscape is significantly more constrained in scale. This data scarcity problem further limits the representational capabilities and generalizability of 3D encoders. Additionally, the inherent scale disparity between 3D encoders and LLMs creates a fundamental architectural imbalance, where the spatial understanding component becomes a performance bottleneck for the entire framework. These scalability issues collectively impede the advancement of 3D language multimodal models,

particularly in tasks requiring fine-grained spatial understanding, generalizing to unseen object categories, or reasoning about complex physical interactions.

## C  IMPLEMENTATION DETAILS

All experimental stages of **Lemon** are conducted on 8 Nvidia H100 GPUs. We employ a consistent training recipe across all model variants as detailed in Table 5.

## D  EXPERIMENTS

### D.1  CONCRETE EXAMPLE

For a point cloud partitioned into $2 \times 3 \times 3 = 18$ patches, the token sequence structure becomes:

```
<pointcloud>
[(0,0,0), (0,0,1), (0,0,2)] <row_sep>
[(0,1,0), (0,1,1), (0,1,2)] <row_sep>
[(0,2,0), (0,2,1), (0,2,2)] <layer_sep>
[(1,0,0), (1,0,1), (1,0,2)] <row_sep>
[(1,1,0), (1,1,1), (1,1,2)] <row_sep>
[(1,2,0), (1,2,1), (1,2,2)]
<pointcloud/>
```

### D.2  ABLATION STUDY ON 3D ENCODERS.

We investigate the impact of different 3D encoder training strategies on overall model performance. Following common practice in 2D/3D LMMs, we compare frozen encoder weights against end-to-end fine-tuning approaches.

| Method | Object Captioning (SimCSE) | Scene Spatial QA |
|---|---|---|
| **Lemon** with Frozen PointBERT | 40.51 | 38.24 |
| **Lemon** with Frozen ReCon++ | 45.37 | 41.32 |
| **Lemon** with Frozen PointNet++ | 41.89 | 32.25 |
| **Lemon** with Fine-tuned ReCon++ | 44.28 | 34.42 |
| **Lemon** with Fine-tuned PointNet++ | 38.73 | 29.58 |
| **Lemon**-7B | **53.59** | **53.45** |

Table 6: Ablation study on different encoder and training strategies.

As presented in Table 6, incorporating external encoders results in suboptimal performance compared to our unified architecture. While advanced encoders like ReCon++ used by ShapeLLM (Qi et al., 2024) provide better representations than PointNet++ (improving object captioning from 41.89 to 45.37), they still significantly lag behind **Lemon**. This performance gap likely stems from the limited generalization capability of pre-trained encoders, which are typically trained on specific narrow domains (e.g., synthetic ShapeNet objects). Furthermore, end-to-end fine-tuning of PointNet++ leads to performance degradation, likely due to training instability when jointly training heterogeneous modules. These results confirm that our unified transformer approach, which treats 3D patches as native tokens, offers a more effective solution for 3D-language modeling than adapting external 3D encoders.

### D.3  ABLATION STUDY ON POINT CLOUD PATCHES

To determine the optimal point cloud patchification strategy, we conduct ablation studies on two critical hyperparameters: the number of points per patch and the maximum number of patches and present the results on captioning performance measured by SimCSE scores.

**Point Counts Per Patch Analysis.** As shown in Figure 6a, we evaluate different point counts per patch. The results demonstrate a consistent performance improvement as the point count increases, with higher point density achieving the best captioning performance. This trend indicates that

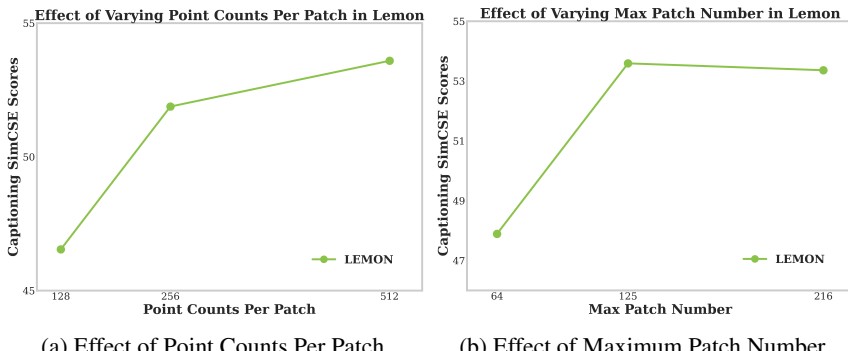

(a) Effect of Point Counts Per Patch.    (b) Effect of Maximum Patch Number.

denser point representation within each patch provides richer spatial information, enabling better understanding of local geometric structures and subsequently improving language generation quality.

**Maximum Patch Number Analysis.**    Figure 6b explores the effect of varying the maximum number of patches. The performance initially increases with more patches and reaches an optimal point, with performance remaining relatively stable at higher patch numbers. This suggests an optimal balance between spatial coverage and sequence length efficiency—too few patches result in insufficient spatial detail, while beyond a certain threshold, additional patches provide diminishing returns in terms of performance gains.

Based on these ablation results, we adopt our final configuration for **Lemon**, which provides the optimal trade-off between spatial representation quality and computational efficiency.

### D.4    IMPACT OF TOKENIZATION AND SPATIAL ORDERING

We investigate the effectiveness of our tokenization strategy by comparing it against alternative mechanisms. We evaluate two major categories: (1) **FPS-based Sampling** (PointBERT-style), which generates spatially discontinuous tokens; and (2) **Space-Filling Curves** (Hilbert/Z-order), which preserve mathematical locality but introduce complex traversal paths.

Table 7: Ablation study on tokenization strategies. Our Z→Y→X strategy achieves the best performance.

| Tokenization Strategy | Spatial Ordering | Object Captioning (SimCSE) | Scene Spatial QA (Accuracy) |
|---|---|---|---|
| FPS-based Sampling (PointBERT-style) | Discontinuous (FPS) | 43.15 | 35.20 |
| Dynamic Patchification (Hilbert SFC) | Structured (Curve) | 47.10 | 49.50 |
| Dynamic Patchification (Z-order SFC) | Structured (Curve) | 48.80 | 48.20 |
| **Dynamic Patchification (Ours)** | **Structured (Z→Y→X)** | **53.59** | **53.45** |

As shown in Table 7, our structured **Z→Y→X** ordering achieves superior performance. The FPS-based baseline suffers significantly (dropping to 35.20% in Spatial QA), confirming that spatially discontinuous sequences disrupt the autoregressive modeling capability of LLMs. Furthermore, while Space-Filling Curves (SFCs) preserve mathematical locality, our simple Z→Y→X ordering outperforms them. This is likely because our strategy aligns with the **gravitational and semantic hierarchy** of indoor scenes (e.g., floor → table → object), offering a logical flow consistent with human descriptions. Additionally, this approach is similar to the **grid-based patch flattening** strategy widely adopted in 2D LMMs (e.g., LLaVA-NEXT (Liu et al., 2023a)), producing a structured sequence that is easier for the LLM to interpret compared to the convoluted traversal paths of Hilbert curves.

### D.5    ABLATION STUDY ON SPATIAL SEPARATOR TOKENS.

Our patchification strategy relies on specialized separator tokens (`<layer_sep>`, `<row_sep>`) to preserve the hierarchical 3D spatial structure (Z→Y→X) within the flattened 1D token sequence. To

quantify their contribution, we conducted an ablation study by removing these special tokens and training the model using only the raw sequence of point patch embeddings.

Table 8 presents the comparison results. The removal of spatial separator tokens leads to significant performance degradation across all tasks. Notably, the decline is most pronounced in Scene Spatial QA, compared to Object Captioning. This disparity indicates that while the model can still recognize object features from local patches, the explicit spatial structure provided by separator tokens is indispensable for complex spatial reasoning tasks, such as understanding relative positions and scene layouts.

Table 8: **Ablation on Spatial Separator Tokens.** The results demonstrate that explicit spatial separators are critical for preserving geometric hierarchy, especially for complex scene-level spatial reasoning.

| Model Variant | Object Captioning (SimCSE) | Scene Spatial QA (GPT-4) |
|---|---|---|
| **Lemon** w/o Spatial Tokens | 45.84 | 40.23 |
| **Lemon (Full)** | **53.59** | **53.45** |

## D.6 ADDITIONAL RESULTS

**Evaluation on Standard Captioning Metrics.** To address potential concerns regarding evaluation bias and ensure comprehensive assessment, we additionally report standard lexical metrics including BLEU-1 and ROUGE-L. However, consistent with findings in PointLLM (Xu et al., 2024) and ShapeLLM (Qi et al., 2024), we observe that these n-gram-based metrics are often unreliable for open-ended 3D object captioning, as they tend to penalize semantically correct but structurally diverse descriptions.

Table 9: Comparison on standard lexical metrics. **Lemon** achieves competitive performance on traditional n-gram metrics while maintaining a significant lead in semantic evaluation (GPT-4).

| Model | BLEU-1 | ROUGE-L | GPT-4 Score |
|---|---|---|---|
| PointLLM | 17.09 | **20.99** | 44.27 |
| 3D-LLM | 16.91 | 19.48 | 33.42 |
| **Lemon (Ours)** | **17.34** | 20.86 | **50.83** |

As shown in Table 9, **Lemon** achieves competitive results on lexical metrics (ranking first on BLEU-1 and comparable on ROUGE-L) while significantly outperforming baselines on the GPT-4 score. This indicates that while **Lemon** generates diverse textual descriptions that may slightly deviate from ground-truth n-grams, it captures the semantic essence of 3D objects more accurately than existing methods.

**Evaluation on other spatial benchmarks.** We further conduct zero-shot evaluations on established spatial reasoning benchmarks, including MSQA (Linghu et al., 2024) and Beacon3D (Huang et al., 2025), while performing fine-tuned evaluations on ScanQA (Azuma et al., 2022) and SQA3D (Ma et al., 2022) for 2 epochs. As shown in Table 10, **Lemon** achieves superior performance across most metrics among point-cloud-based methods. Specifically, **Lemon** establishes new state-of-the-art results on ScanQA and SQA3D, outperforming recent strong baselines such as Inst3D-LMM and Chat-Scene. Furthermore, on fine-grained diagnosis benchmarks like MSQA and Beacon3D, our model maintains robust performance, demonstrating its exceptional generalization capacity in handling diverse spatial understanding tasks without requiring task-specific architectural modifications.

The results demonstrate consistent performance advantages across multiple established benchmarks, validating the effectiveness and generalization capability of our approach.

**Performance under Challenging Conditions.** We evaluate our model's robustness on sparse and noisy point cloud benchmarks to assess practical applicability. **Lemon** maintains consistent performance even under challenging conditions, benefiting from our large-scale pretraining dataset, which includes point clouds with varying densities.

Table 10: Comparison on spatial reasoning benchmarks. We report CIDEr (C), BLEU-4 (B-4), ROUGE-L (R), and METEOR (M) for ScanQA, and EM & EM-Recall (EM-R) for SQA3D. **Lemon** achieves state-of-the-art performance on most metrics. * indicates zero-shot evaluation.

| Model | ScanQA (val) | | | | SQA3D (test) | | MSQA* | Beacon3D* |
| | C | B-4 | R | M | EM | EM-R | Score | Case |
|---|---|---|---|---|---|---|---|---|
| 3D-LLM (Hong et al., 2023) | 69.4 | 12.0 | 35.2 | 14.8 | 49.8 | - | - | - |
| LSceneLLM (Zhi et al., 2024) | 80.0 | 12.0 | - | - | 54.2 | - | 11.7 | - |
| 3D-VisTA (Zhu et al., 2023) | 72.9 | 13.1 | 42.7 | 13.9 | 48.5 | - | - | 43.2 |
| SceneVerse (Jia et al., 2024) | - | - | - | - | 49.9 | - | - | 40.5 |
| LEO (Huang et al., 2023) | 80.0 | 11.5 | 39.3 | 16.2 | 50.0 | 53.7 | 7.84 | 45.2 |
| Chat-Scene (Huang et al., 2024) | 87.7 | 14.3 | 41.6 | 18.0 | 54.6 | 57.5 | - | **49.8** |
| Inst3D-LMM (Yu et al., 2025) | - | 14.9 | - | - | - | - | - | - |
| **Lemon (Ours)** | **90.5** | **15.4** | **45.1** | **20.3** | **59.4** | **63.0** | **10.68** | 46.2 |

| Condition | Embodied Object QA | Scene Spatial QA (Non-binary) |
|---|---|---|
| Original | 57.22 | 53.45 |
| Noisy ($\sigma = 0.01$) | 55.86 | 50.92 |
| Sparse (50% sampling) | 53.71 | 49.38 |

Table 11: Robustness evaluation under noisy and sparse point cloud conditions.

## D.7 EVALUATION ON 3D VISUAL GROUNDING

To further validate **Lemon**'s fine-grained spatial localization capabilities beyond QA and captioning, we conducted additional experiments on the 3D visual grounding task. We utilized the widely adopted **ScanRefer** benchmark (Chen et al., 2020), which requires the model to localize a specific object in a 3D scene given a natural language description.

Following standard protocols established by baselines, we fine-tuned **Lemon** on the ScanRefer training set and evaluated performance using the Acc@0.5 metric (accuracy of bounding box prediction with IoU $\geq 0.5$).

Table 12: Performance comparison on the ScanRefer validation set (Acc@0.5). **Lemon** achieves competitive localization performance compared to specialized grounding models, demonstrating strong spatial-semantic alignment.

| Model | ScanRefer Acc@0.5 |
|---|---|
| ScanRefer (Chen et al., 2020) | 24.3 |
| 3D-VisTA (Zhu et al., 2023) | 45.8 |
| GPS (Jia et al., 2024) | 48.1 |
| Chat-Scene (Huang et al., 2024) | **50.2** |
| **Lemon (Ours)** | 48.0 |

As shown in Table 12, **Lemon** achieves an accuracy of **48.0%**, which is highly competitive with strong baselines such as GPS (48.1%) and significantly outperforms 3D-VisTA (45.8%). Notably, **Lemon** achieves this performance without incorporating large-scale grounding datasets during the pre-training stage, relying instead on the robust spatial representations learned through our unified architecture. This result confirms that **Lemon** possesses precise 3D localization capabilities essential for tasks such as detection and referring expression comprehension.

## D.8 COMPUTATIONAL EFFICIENCY AND SCALABILITY

We provide a comprehensive analysis of the computational efficiency of **Lemon**, covering training cost, inference latency, and parameter efficiency.

**Training Efficiency.** Our three-stage training curriculum is highly efficient, completing in a total of **78 hours** on 8×H100 GPUs (Stage 1: 48h, Stage 2: 24h, Stage 3: 6h). This rapid convergence is facilitated by our unified architecture, which avoids the instability often associated with jointly optimizing separate 3D encoders.

**Inference Latency.**   We compare the inference speed of **Lemon** against state-of-the-art 2D and 3D multimodal models. As shown in Table 13, **Lemon** achieves superior latency (**0.052s** per token generation step). This speed advantage stems from our encoder-free design, which eliminates the heavy forward pass of external 3D backbones.

Table 13: Inference latency comparison. Times are measured as the average per-token generation latency on a single H100 GPU (Input points $\approx$ 16k).

| Model | Backbone Size | Inference Time (s) |
|---|---|---|
| 3D-LLM | 7B | 0.0762 |
| ShapeLLM | 7B | 0.0745 |
| LLaVA-1.5 | 13B | 0.0672 |
| Qwen2.5-VL | 7B | 0.0588 |
| **Lemon (Ours)** | **7B** | **0.0520** |

**Detailed Compute Breakdown.**   To address concerns regarding the overhead of our Dynamic Patchification (which involves sorting and sampling), we analyze the per-module latency in Table 14. Notably, the visual processing stage accounts for only a small fraction of the total latency. While global FPS can be computationally expensive, our strategy performs **hierarchical spatial sorting first**, which is highly efficient ($O(N \log N)$). FPS is then selectively applied only to local oversized patches, avoiding the quadratic complexity of global sampling on the entire scene.

Table 14: Per-module compute and latency statistics for **Lemon** (Qwen2.5-7B backbone, batch size = 1, FP16, Input Points $\approx$ 16k). The visual processing overhead is minor compared to the LLM backbone.

| Module | Params (M) | FLOPs (G) | Latency / Memory | | |
|---|---|---|---|---|---|
| | | | **Single H100** | **8×H100** | **Memory (GB)** |
| Patchification (Sort + FPS) | 0 | 3.0 | 12.5 / 15.0 ms | 10.5 / 12.0 ms | 0.8 |
| Linear Projector | 6.3 | 1.0 | 2.5 / 3.5 ms | 2.0 / 3.0 ms | 0.5 |
| LLM Backbone (Qwen2.5-7B) | 7610 | 520 | 64.0 / 78.0 ms | 37.5 / 46.0 ms | 14.5 |
| **Total** | **7616.3** | **524** | **79.0 / 96.5 ms** | **50.0 / 61.0 ms** | **15.2** |

**Parameter Efficiency.**   I n Table 15, we compare the architectural complexity. While ShapeLLM and PointLLM fall under the same "7B" category, they require loading external 3D encoders (e.g., ShapeLLM uses a heavy ReCon++ Large encoder with $\sim$500M parameters) in addition to the LLM. In contrast, **Lemon** achieves a streamlined design with **zero** encoder parameters, integrating 3D processing directly into the LLM.

Table 15: Detailed comparison of architectural components and trainable parameters. **Lemon** achieves a streamlined design by removing the standalone 3D encoder.

| Model | 3D Encoder | | Projector | LLM Backbone |
|---|---|---|---|---|
| | **Type** | **Params** | | |
| PointLLM-7B | PointBERT | $\sim$40M | Linear | LLaMA-2 7B ($\sim$7.3B) |
| ShapeLLM-7B | ReCon++-L | $\sim$300M | MLP | Vicuna 7B ($\sim$7.5B) |
| **Lemon-7B** | **None** | **0** | **Linear** | **Qwen2.5-7B (7.2B)** |

# E   DATASET AND BENCHMARK

## E.1   DETAILS OF EMBODIED 3D SPATIAL QA SET

To evaluate the model's capability in handling complex spatial reasoning tasks required for embodied agents, we constructed a specialized test set comprising 100 challenging samples. We sourced the 3D

Table 16: Detailed statistics of the 100 Challenging 3D Spatial QA set. The dataset is manually curated to cover diverse aspects of embodied spatial reasoning.

| Task Category | Focus & Example | Count |
|---|---|---|
| Navigability Analysis | Passability checks (e.g., "Can a robot pass through...?") | 30 |
| Precise Distance Estimation | Relative distance comparison (e.g., "Closer to A or B?") | 25 |
| Collision & Interaction | Physics/Safety (e.g., "Will it hit the table if fell?") | 20 |
| Spatial Relations | Complex positioning (e.g., "Behind/Next to under occlusion") | 25 |
| **Total** | | **100** |

Table 17: Composition of the object evaluation set. We explicitly include diverse real-world scanned datasets to verify robustness against noise and occlusion.

| Dataset | Source Type | Characteristics | Count |
|---|---|---|---|
| ShapeNet | Synthetic | Clean CAD models | 400 |
| Structured3D | Synthetic | Photorealistic simulation | 400 |
| ScanNet | **Real-world** | RGB-D Scans (Indoor) | 400 |
| 3RScan | **Real-world** | Temporal Scans | 400 |
| ARKitScenes | **Real-world** | Mobile Lidar/RGB | 400 |
| **Total** | | **Mixed Domains** | **2000** |

scenes from the 3D-GRAND dataset, specifically selecting dense and cluttered indoor environments such as bathrooms, kitchens, living rooms, and bedrooms. These scenes were chosen to provide rich geometric contexts where spatial relationships are intricate and require precise 3D understanding beyond simple object recognition.

Based on these selected point clouds, we engaged human experts to manually design Question-Answer pairs focused on embodied interaction scenarios. Unlike general captions, these questions are specifically tailored to test rigorous spatial reasoning capabilities. As detailed in Table 16, the dataset covers specific embodied tasks including precise distance estimation, navigability analysis, and collision avoidance. By incorporating these manually verified, high-difficulty cases, this set serves as a robust benchmark for assessing fine-grained spatial intelligence in realistic 3D environments.

### E.2 DETAILS OF OBJECT EVALUATION DATASETS

As shown in Table 17, our 3D object evaluation set covers both synthetic environments and challenging real-world scanned scenes (e.g., ScanNet, ARKitScenes), ensuring a balanced assessment of the model's robustness.

## F EVALUATION PROMPTS

We provide the specific prompts used for our LLM-as-judge evaluations to ensure reproducibility. Following the protocols in previous works, the prompts for Object Recognition, Object Captioning, and Embodied Object QA are adopted from ShapeLLM (Qi et al., 2024). For the Scene Spatial Awareness QA, we reference the evaluation design from 3D-GRAND (Yang et al., 2024a), tailoring the prompt to cover diverse aspects including Navigability Analysis, Precise Distance Estimation, Collision & Interaction, and Spatial Relations.

**Object Recognition Evaluation Prompt**

```
Analyze two sentences and determine if they're referring to the
    same general object or concept, focusing on the type of object,
    not attributes such as color, size, or shape. Respond with 'T'
    if they refer to the same thing and 'F' if not. Also, provide a
    brief rationale (no more than 20 words) for your judgment.

Example:
Input: 1. Spiral staircase that goes from a ground floor. 2. This
    is a 3D model of wooden stairs in light brown
Output: T\#Both refer to a staircase.

Now, analyze the following:
Input: 1. \{ground\_truth\} 2. \{model\_output\}
Output:
```

---

**Object Captioning Evaluation Prompt**

```
Evaluate a model-generated caption against a human-generated
    caption (ground truth) for a 3D model. Identify the aspects
    mentioned in the human caption and calculate the percentage of
    these aspects correctly mentioned or partially matched in the
    model caption. Score from 0 to 100, where each aspect
    contributes equally to the score. Consider similar concepts for
    partial score.

Provide your score (0-100) and a short justification (less than 15
    words) in the format of 'score\#reason'

Example:
Human: A white brown skeleton
Model: This is a 3D model of a small, cartoon-like robot. It has a
    spherical body and is covered in a layer of white dust.
Output: 50\#mention white; skeleton and robot have similar
    appearence.

Now score the following:
Human: \{ground\_truth\}
Model: \{model\_output\}
Output:
```

**Embodied Object QA Evaluation Prompt (3D-MM-Vet)**

```
Now I will give you a question, the type of the question, an answer
    from model, and an answer from label.
All you need to do is focus on these two answers and figure out
    whether they are saying the same thing about the specific type
    of question.
Your response should only be a confidence score ranging from 0 to
    100.
Remember the confidence score is to evaluate how much two answers
    are describing the same thing.
Your response confidence score should follow the scoring standard
    of the prompt I gave.
Firstly I will give you several question \& answer pairs as long as
    their confidence score:

question1: How many oranges will there be if 1/3 of them are
    removed?
question type: Knowledge
answer from model: There will be 6 left.
answer from label: As there are 9 oranges in total, there will be 6
    oranges left if 1/3 of them are removed.
confidence score: 100

question2: What is this object?
question type: General Visual Recognition
answer from model: This is a bathtub
answer from label: This is a dirty bathtub.
confidence score: 80

question3: What is this object?
question type: General Visual Recognition
answer from model: This is a bottle of water
answer from label: This is a bottle of oil
confidence score: 50

question4: What is holding in this boy's right hand?
question type: Spatial Recognition
answer from model: He is holding a white cup in his right hand.
answer from label: He is holding a sword in his right hand.
confidence score: 0

Next, I will give you the elements:
question: \{question\},
question type: \{type\},
answer from model: \{model\_output\},
answer from label: \{ground\_truth\}.
Please remember, while outputting the confidence score, do not
    include any words, just the number.
```

### Scene Spatial Awareness QA Evaluation Prompt

Now I will give you a question about a 3D scene, the type of the
    question, an answer from the model, and an answer from the label
    .
All you need to do is focus on these two answers and determine
    whether the model's answer conveys the same spatial information
    or reasoning as the label, given the specific question type.
Your response should only be a confidence score ranging from 0 to
    100.
Remember the confidence score is to evaluate the accuracy of the
    spatial understanding and reasoning.
Your response confidence score should follow the scoring standard
    of the prompt I gave.
Firstly I will give you several question \& answer pairs as long as
     their confidence score:

question1: Is the coffee table closer to the sofa or the TV stand?
question type: Spatial Relations
answer from model: It is closer to the sofa.
answer from label: The table is positioned right in front of the
    sofa, far from the TV.
confidence score: 100

question2: Can a robot vacuum pass between the bed and the wall?
question type: Navigability Analysis
answer from model: Yes, there is plenty of space.
answer from label: No, the gap is too narrow for a robot to
    navigate.
confidence score: 0

question3: What is the distance between the ceiling lamp and the
    floor?
question type: Precise Distance Estimation
answer from model: It is about 2 meters.
answer from label: The lamp hangs approximately 2.5 meters above
    the ground.
confidence score: 80

question4: If I open the wardrobe door, will it hit the bedside
    table?
question type: Collision \& Interaction
answer from model: No, there is enough clearance.
answer from label: Yes, the door swing radius intersects with the
    table.
confidence score: 0

Next, I will give you the elements:
question: \{question\},
question type: \{type\},
answer from model: \{model\_output\},
answer from label: \{ground\_truth\}.
Please remember, while outputting the confidence score, do not
    include any words, just the number.

