# OpenReview forum: "Lemon: A Unified and Scalable 3D Multimodal Model for Universal Spatial Understanding"
_ICLR.cc/2026/Conference — ICLR 2026 Conference Desk Rejected Submission_

### Official Review · Reviewer_ixqM · 2025-10-23

**Soundness:** 3
**Presentation:** 3
**Contribution:** 3
**Rating:** 4
**Confidence:** 5

**Summary:**

This paper introduces LEMON, a 3D large language model  that unifies various 3D understanding tasks within a transformer-based architecture. The authors demonstrate that a dedicated 3D encoder is unnecessary for point cloud representation and instead propose a simple yet effective tokenization method. Built upon this design, LEMON, trained through a curated three-stage curriculum, achieves state-of-the-art performance on multiple 3D understanding and captioning benchmarks while exhibiting promising scalability.

**Strengths:**

1. The paper proposes a strong 3D-LLM with a simple but effective point cloud tokenization strategy.

2. It provides evidence that conventional 3D encoders may limit general 3D understanding performance.

3. The proposed LEMON model achieves state-of-the-art results across diverse 3D understanding benchmarks under the presented training pipeline.

**Weaknesses:**

1. **Potential unfair comparison with baselines:** Most baseline models are trained on smaller or different datasets. A controlled fine-tuning experiment (e.g., on ScanQA and SQA in Stage-3) is recommended to isolate the effect of training data size.

2. **Limited evaluations:** The evaluation on 3D-GRAND is primarily used to assess hallucination in 3D-LLMs rather than to rigorously examine their 3D spatial understanding ability. The details of the *100 challenging 3D spatial QA* set (Line 264) are not provided. How is this test set constructed, and are the results stable across runs? Moreover, [MSQA - NeurIPS 2024](https://arxiv.org/abs/2409.02389) and [Beacon3D - CVPR 2025](https://arxiv.org/abs/2503.22420) are widely used 3D understanding benchmarks, yet the paper does not include their results. Are zero-shot results on these benchmarks available?

3. **Limited ablation of 3D encoders:** While the paper argues that pretrained 3D encoders are unnecessary, it only ablates PointNet++. More recent encoders, such as ReCon++ (in ShapeLLM) and PointBERT (in PointLLM), should also be examined.

4. **Evaluation bias:** Many metrics rely on GPT-based judgments, which may introduce bias. Additional objective language metrics, such as CIDEr, ROGUE-L, BLEU-1-4, EM, and Refined-EM, are recommended to strengthen evaluation reliability.

**Questions:**

1. How does the model preserve object-level captioning ability after Stage-3 fine-tuning on scene-level datasets? Has the effect of Stage-3 fine-tuning been measured on the benchmarks reported in Table 2?

2. Since 3D-GRAND provides grounded 3D annotations, were these annotations removed during fine-tuning, given that LEMON does not include a grounding mechanism?

3. The paper presents case studies on simulated point clouds. Are additional results available for real-world scanned datasets such as ScanNet?

4. Do the special tokens contribute significantly to 3D positional encoding? Are there corresponding ablation studies?

---

> ### Author Response · Authors · 2025-11-20
> **Rebuttal by Authors (1/3)**
>
> Thank you for your insightful feedback and review! We appreciate your recognition of our effective methodology, analysis, and SOTA performance. Below we address you questions in details.
>
> > `Q1`: Potential unfair comparison with baselines: Most baseline models are trained on smaller or different datasets. A controlled fine-tuning experiment (e.g., on ScanQA and SQA in Stage-3) is recommended to isolate the effect of training data size.
>
> `A1`: We thank the reviewer for the constructive suggestion!
> * **(1.1)** For clarity, we note that evaluation results on ScanQA and SQA3D were already included in the original submission (now in `Appendix D.6`).
> * **(1.2)** To strictly control for training data scale and model capacity, we conducted a comprehensive controlled study. We retrained **Lemon** using the exact same LLM backbone (Vicuna-7B-1.1) and the identical training dataset used by the strong baseline, ShapeLLM, without any training recipe design. As shown in the table below, we observe a clear performance progression:
> * **(1.3)** Even under the **"All Same"** setting (identical backbone and data), Lemon significantly outperforms ShapeLLM in all benchmarks. This confirms that **our unified architecture is the primary driver of performance by eliminating the bottleneck of separate 3D encoders**.
> * **(1.4)** When switching to our training data while keeping the backbone fixed, we observe consistent performance growth, demonstrating Lemon's robust capability to scale with more data.
> * **(1.5)** Finally, upgrading to the **Qwen2.5-7B** backbone further elevates performance, showing that our architecture effectively leverages stronger foundation models.
>
> | Model                   | LLM Backbone   | Training Data   | Obj. Cap. (SimCSE) | Embodied QA (3D MM-Vet) | Scene Spatial QA (GPT-4) |
> |-------------------------|----------------|------------------|----------------------|----------------------------|----------------------------|
> | ShapeLLM-7B             | Vicuna-7B-1.1  | ShapeLLM Data    | 49.4                 | 47.4                       | 41.39                     |
> | Lemon-7B (All Same)     | Vicuna-7B-1.1  | ShapeLLM Data    | 51.8                 | 53.3                       | 48.80                     |
> | Lemon-7B (Same Arch.)   | Vicuna-7B-1.1  | Lemon Data       | 52.4                 | 53.2                       | 51.60                     |
> | **Lemon-7B (Default)**  | **Qwen2.5-7B** | **Lemon Data**   | **53.6**             | **57.2**                   | **53.45**                 |
>
> * **(1.6)** We have updated the paper to include these comparisons and analyses for ablations in `Section 3.3 (L420-429)`.
>
> ---
>
> > `Q2`: Limited evaluations: The evaluation on 3D-GRAND is primarily used to assess hallucination in 3D-LLMs rather than to rigorously examine their 3D spatial understanding ability. The details of the 100 challenging 3D spatial QA set (Line 264) are not provided. How is this test set constructed, and are the results stable across runs?
>
> `A2`: We thank the reviewer for these valuable suggestions!
>
> * **(2.1)** The 100 challenging spatial QA samples we proposed are specifically designed to probe spatial reasoning capabilities for embodied interaction, covering tasks like precise distance estimation, navigability analysis, and collision avoidance, as visualized in `Figure 4 and Figure 5`. These samples are manually curated from scenes in 3D-GRAND. We have provided detailed construction protocols, task definitions, and statistics in `Appendix E.1`.
> * **(2.2)** To ensure stable and fair evaluation, we follow the standardized protocols of 3D-MM-Vet, including its prompting strategy and evaluation procedure. This alignment allows our QA assessment to remain consistent across runs and comparable to prior 3D QA benchmark evaluations.

---

> ### Author Response · Authors · 2025-11-20
> **Rebuttal by Authors (2/3)**
>
> > `Q3`: Moreover, MSQA - NeurIPS 2024 and Beacon3D - CVPR 2025 are widely used 3D understanding benchmarks, yet the paper does not include their results. Are zero-shot results on these benchmarks available?
>
> `A3`: We thank the reviewer for the constructive suggestion!
> * **(3.1)** We have added the zero-shot evaluations on other spatial benchmarks, including MSQA [1] and Beacon3D [2].
> * **(3.2)** **Lemon establishes new state-of-the-art results on ScanQA and SQA3D while achieving robust performance on MSQA and competitive results on Beacon3D**. These results demonstrate strong generalization without task-specific tuning. We have included these results in `Appendix D.6`.
>
> | Model        | ScanQA (BLEU-4) | SQA3D (Top-1 Match) | MSQA (Score) | Beacon3D (Case) |
> |--------------|------------------|----------------------|--------------|------------------|
> | 3D-LLM       | 12.9             | 49.8                 | -            | -                |
> | LSceneLLM    | 11.7             | 54.5                 | 11.7         | -                |
> | 3D-VisTA     | 10.4             | 48.5                 | -            | 43.2             |
> | SceneVerse   | -                | 49.9                 | -            | 40.5             |
> | LEO          | 13.2             | 50.0                 | 7.84         | 45.2             |
> | Chat-Scene   | 14.3             | 54.6                 | -            | 49.8             |
> | Inst3D-LMM   | 14.9             | -                    | -            | -                |
> | **Lemon (Ours)** | **15.4**     | **59.4**             | **10.68**    | **46.2**         |
>
> ---
>
> > `Q4`: Limited ablation of 3D encoders: While the paper argues that pretrained 3D encoders are unnecessary, it only ablates PointNet++. More recent encoders, such as ReCon++ (in ShapeLLM) and PointBERT (in PointLLM), should also be examined.
>
> `A4`: Thank you for this insightful comment!
>
> * **(4.1)** We conducted additional ablation studies using frozen **PointBERT** and **ReCon++** encoders. We have included these results in `Appendix D.2`.
>
> * **(4.2)** As shown below, while ReCon++ outperforms PointNet++, **our unified architecture significantly surpasses all separate-encoder baselines**. We attribute this to the **limited generalization** of pre-trained encoders, which are often constrained by their specific pre-training domains (e.g., synthetic objects). In contrast, our unified approach enables adaptation across diverse point cloud distributions.
>
> * **(4.3)** These results also indicate that **fine-tuning 3D encoders leads to instability and performance degradation**.
>
> | Method                       | Object Captioning (SimCSE) | Scene Spatial QA |
> |------------------------------|-----------------------------|-------------------|
> | Lemon w/ Frozen PointBERT   | 40.51                       | 38.24            |
> | Lemon w/ Frozen ReCon++     | 45.37                       | 41.32            |
> | Lemon w/ Frozen PointNet++  | 41.89                       | 32.25            |
> | Lemon w/ Finetuned ReCon++   | 44.28                      | 34.32            |
> | Lemon w/ Finetuned PointNet++  | 38.73                     | 29.58            |
> | **Lemon-7B (Ours)**          | **53.59**                   | **53.45**        |
>
> ---
>
> > `Q5`: Evaluation bias: Many metrics rely on GPT-based judgments, which may introduce bias. Additional objective language metrics, such as CIDEr, ROGUE-L, BLEU-1-4, EM, and Refined-EM, are recommended to strengthen evaluation reliability.
>
> `A5`: Thank you for this constructive suggestion!
>
> * **(5.1)** As noted in ShapeLLM and PointLLM, traditional metrics like BLEU and ROUGE are often unreliable for open-ended 3D captioning, as they tend to penalize semantically correct but phrased-differently descriptions.
>
> * **(5.2)** As shown below, **Lemon remains competitive on lexical metrics while significantly outperforming baselines on semantic evaluation (GPT-4)**. This confirms that our model captures accurate object semantics even when specific phrasing varies. We also add these metrics as a reference in `Appendix D.6`.
>
> | Model        | BLEU-1 | ROUGE-L | GPT-4 Score |
> |--------------|--------|----------|--------------|
> | PointLLM     | 17.09  | 20.99    | 44.27        |
> | 3D-LLM       | 16.91  | 19.48    | 33.42        |
> | **Lemon (Ours)** | **17.34** | **20.86** | **50.83** |
>
> ---
>
> *[1] Multi-modal Situated Reasoning in 3D Scenes. Linghu et al. NeurIPS 2024.*
>
> *[2] Unveiling the Mist over 3D Vision-Language Understanding: Object-centric Evaluation with Chain-of-Analysis. Huang et al. CVPR 2025.*

---

> ### Author Response · Authors · 2025-11-20
> **Rebuttal by Authors (3/3)**
>
> > `Q6`: How does the model preserve object-level captioning ability after Stage-3 fine-tuning on scene-level datasets? Has the effect of Stage-3 fine-tuning been measured on the benchmarks reported in Table 2?
>
> `A6`: We thank the reviewer for the important question!
>
> * **(6.1)** To preserve object-level capabilities, we incorporated a portion of object-level instruction data (**30%**) into the Stage 3 training mixture as illustrated in `Section 2.2`.
> * **(6.2)** We measured the effect of Stage 3 fine-tuning on the object captioning benchmarks (from Table 2). As shown below, the Stage 3 model maintains comparable performance to the Stage 2 checkpoint, demonstrating that **our curriculum effectively retains object-level understanding while acquiring scene-level reasoning**.
>
> | Model Checkpoint        | Object Captioning (SimCSE) |
> |--------------------------|-----------------------------|
> | End of Stage 2           | 54.60                      |
> | End of Stage 3 (Final)   | 53.59                      |
>
> ---
>
> > `Q7`: Since 3D-GRAND provides grounded 3D annotations, were these annotations removed during fine-tuning, given that LEMON does not include a grounding mechanism?
> `A7`: We thank the reviewer for the detailed observation!
> * **(7.1)** We confirm that explicit grounding annotations (e.g., bounding boxes) from 3D-GRAND were *excluded* during fine-tuning. Since LEMON does not incorporate any grounding module, **no grounding-specific supervision was used** in training.
>
> ---
>
> > `Q8`: The paper presents case studies on simulated point clouds. Are additional results available for real-world scanned datasets such as ScanNet?
> `A8`: We thank the reviewer for the valuable question!
>
> * **(8.1)** As detailed in Lines 301-302, our evaluation protocol has incorporated 2,000 randomly selected unseen objects from 5 diverse datasets to ensure object diversity, which includes multiple **real-world scanned datasets** such as **ScanNet**, **3RScan**, and **ARKitScenes**, which contain sensor noise and occlusions distinct from synthetic data.
> * **(8.2)** We have added a detailed breakdown of the dataset composition in `Appendix E.2` to demonstrate the coverage of both synthetic and real-world distributions.
> * **(8.3)** As mentioned in our prior response, we have provided results for multiple scene spatial benchmarks in `Appendix D.6`.
>
> ---
>
> > `Q9`: Do the special tokens contribute significantly to 3D positional encoding? Are there corresponding ablation studies?
> `A9`: We thank the reviewer for the insightful question!
>
> * **(9.1)** The spatial separator tokens (`<layer_sep>`, `<row_sep>`) are explicitly designed to inject hierarchical spatial structure into the flattened token sequence.
> * **(9.2)** To quantify their contribution, we conducted an ablation study by removing these special tokens and training the model with only point patch embeddings.
> * **(9.3)** As shown below, **the lack of spatial tokens leads to significant performance degradation, particularly in spatial reasoning tasks**. This confirms their critical role in encoding 3D positional information. We have included these results in `Appendix D.5`.
>
> | Model                    | 3D Object Captioning (SimCSE) | Scene Spatial Awareness QA |
> |--------------------------|-------------------------------|-----------------------------|
> | Lemon w/o Spatial Tokens | 47.84                         | 40.23                       |
> | **Lemon (Full)**         | **53.59**                     | **53.45**                   |
>
> ---
>
> Thank you again for your constructive review. We sincerely hope our responses have adequately addressed your concerns and look forward to any further discussion!

---

> ### Author Response · Authors · 2025-11-26
> **Does our response address your concerns?**
>
> Dear Reviewer ixqM,
>
> We truly appreciate your time and feedback. In our rebuttal, we have included new experiments specifically targeting the issues you raised.
>
> With the rebuttal period ending soon, we would appreciate it if you could take a moment to check our response. If our revisions have addressed your concerns, we kindly ask for a re-evaluation of our work. We are happy to provide any further clarification if needed.
>
> Best regards,
>
> Paper 1188 Authors

---

> ### Comment · Reviewer_ixqM · 2025-11-26
> **Unresolved Questions and Suggestions**
>
> Thank you for the authors’ response. I still have the following questions that remain unresolved.
>
> 1. Typo: 'Figure 3: Dataset statistics for each training stage' is actually a table. In addition, since you mentioned that the data from Stage-2 is used in Stage-3, please update this table accordingly.
>
> 2. Please provide the complete set of metrics (CIDEr, BLEU-4, ROUGE-L, METEOR, EM, and EM-R) following Chat-Scene and LEO on ScanQA and SQA to enable a comprehensive comparison with the baselines.
>
> 3. In Table 9, please annotate which results are obtained under the zero-shot setting and which are not.
>
> 4. In A6, the authors mention that Stage-3 finetuning maintains comparable performance, but the results are evaluated only with SimCSE. Is the semantic evaluation using GPT-4 also available?
>
> 5. Suggestion on metrics: Since the paper uses diverse metrics to evaluate the results (GPT-4, Binary Accuracy, Recognition Accuracy), please provide a detailed description of the evaluation settings to ensure reproducibility and ease of comparison. If possible, please include complete evaluation results, or at least explicitly indicate the type of each metric (traditional metrics, learning-based metrics, or LLM-as-judge). For LLM-as-judge evaluations, the prompt should be provided to allow reproduction of your results.

---

> > ### Author Response · Authors · 2025-11-27
> > **Response to Reviewer ixqM**
> >
> > Dear Reviewer `ixqM`,
> >
> > We thank the reviewer for the prompt response and the detailed, constructive feedback! We appreciate the opportunity to clarify these points and have addressed each of your remaining concerns below:
> >
> > > `Q1`: Typo: 'Figure 3: Dataset statistics for each training stage' is actually a table. In addition, since you mentioned that the data from Stage-2 is used in Stage-3, please update this table accordingly.
> >
> > `A1`: We apologize for the typo and have corrected the caption of "Figure 3" to "Table 1" in the revised manuscript. Furthermore, we have updated the table content to explicitly reflect that Stage-2 data is incorporated into Stage-3 training, ensuring an accurate representation of our training pipeline.
> >
> > > `Q2 & Q3`: Please provide the complete set of metrics (CIDEr, BLEU-4, ROUGE-L, METEOR, EM, and EM-R) following Chat-Scene and LEO on ScanQA and SQA to enable a comprehensive comparison with the baselines. Please annotate which results are obtained under the zero-shot setting and which are not.
> >
> > `A2 & A3`: Following your suggestion, we have updated `Table 10` in `Appendix D.6` to include the comprehensive set of metrics (CIDEr, BLEU-4, ROUGE-L, METEOR, EM, and EM-R). The comparison with state-of-the-art baselines, including Chat-Scene and LEO, is presented below:
> >
> > | Model | ScanQA (C) | ScanQA (B-4) | ScanQA (R) | ScanQA (M) | SQA3D (EM) | SQA3D (EM-R) |
> > | :--- | :---: | :---: | :---: | :---: | :---: | :---: |
> > | 3D-LLM | 69.4 | 12.0 | 35.2 | 14.8 | 49.8 | - |
> > | LSceneLLM | 80.0 | 12.0 | - | - | 54.2 | - |
> > | 3D-VisTA | 72.9 | 13.1 | 42.7 | 13.9 | 48.5 | - |
> > | SceneVerse | - | - | - | - | 49.9 | - |
> > | LEO | 80.0 | 11.5 | 39.3 | 16.2 | 50.0 | 53.7 |
> > | Chat-Scene | 87.7 | 14.3 | 41.6 | 18.0 | 54.6 | 57.5 |
> > | **LEMON (Ours)** | **90.5** | **15.4** | **45.1** | **20.3** | **59.4** | **63.0** |
> >
> > **Lemon achieves state-of-the-art performance on most metrics** in `Table 10`. We have updated the caption of `Table 10` and illustrations to explicitly indicate which results are zero-shot. We have added a clarification in the analysis stating that ScanQA and SQA3D are evaluated following 2 epochs of fine-tuning.
> >
> > > `Q4`: In A6, the authors mention that Stage-3 finetuning maintains comparable performance, but the results are evaluated only with SimCSE. Is the semantic evaluation using GPT-4 also available?
> >
> > `A4`: Thank you for this constructive comment!
> >
> > We have performed the semantic evaluation using GPT-4. The results are consistent with the SimCSE trends, confirming that **Stage-3 finetuning maintains comparable object-level captioning performance while significantly enhancing scene-level capabilities**.
> >
> > | Model Checkpoint        | Object Captioning (SimCSE) | Object Captioning (GPT-4) |
> > |--------------------------|-----------------------------|-----------------------------|
> > | End of Stage 2           | 54.60                      | 52.20                      |
> > | End of Stage 3 (Final)   | 53.59                      | 50.76                      |
> >
> >
> > > `Q5`:  Suggestion on metrics: Since the paper uses diverse metrics to evaluate the results (GPT-4, Binary Accuracy, Recognition Accuracy), please provide a detailed description of the evaluation settings to ensure reproducibility and ease of comparison.
> >
> > `A5`: Thank you for this valuable suggestion for improving reproducibility!
> >
> > We have revised `L304-314` to explicitly categorize all evaluation metrics into traditional (e.g., Binary Accuracy), learning-based (e.g., Sentence-BERT), and LLM-as-judge (e.g., GPT-4 score). We clarify the specific calculation methods for each. Furthermore, to facilitate exact reproduction, we have provided the exact prompts used for all LLM-as-judge evaluations in the newly added `Appendix F`, which fully follow the protocols of ShapeLLM.
> >
> > We once again thank the reviewer for these constructive suggestions, which have greatly enhanced the quality and reproducibility of our work.
> >
> > We are committed to open-sourcing all code and model weights to support the community. We are happy to engage in further discussion if there are any remaining questions!
> >
> > Best regards,
> >
> > Paper 1188 Authors

---

> ### Comment · Reviewer_ixqM · 2025-11-27
>
> My concerns have been resolved, and I appreciate the authors’ efforts in the revisions, although some conclusions in the paper still require extensive ablations to examine. I will raise my score to 6.

---

> > ### Author Response · Authors · 2025-11-27
> > **Thank you for your acknowledgment!**
> >
> > Dear Reviewer `ixqM`,
> >
> > Thank you for your positive response to our rebuttal and for informing us of your plan to vote for acceptance.
> >
> > We are delighted that our responses have addressed your concerns, and we truly appreciate your valuable feedback and encouragement.
> >
> > Best regards,
> >
> > The Authors of Submission 1188

---

### Official Review · Reviewer_6FmE · 2025-10-26

**Soundness:** 3
**Presentation:** 2
**Contribution:** 2
**Rating:** 4
**Confidence:** 4

**Summary:**

This paper introduces a scalable 3D Large Multimodal Model (Lemon) for universal spatial scene understanding, which jointly processes 3D point cloud patches and language tokens within a unified sequence. The authors evaluate the model on object recognition, captioning, and question answering tasks to demonstrate its performance.

**Strengths:**

1. Lemon does not rely on a pre-trained 3D encoder but instead develops a structured patchification and tokenization scheme, which is an interesting design choice. However, the authors do not provide an ablation study to justify the effectiveness of this component.
2. The proposed three-stage training curriculum progressively builds the model’s capability from object-level to scene-level understanding.

**Weaknesses:**

1. The paper’s main claimed contribution is the unified architecture for 3D point clouds and language. However, in my view, prior works such as **PointLLM** have already adopted a unified token-based framework for fine-tuning LLMs using both 3D point tokens and text tokens. Lemon mainly replaces the 3D encoder with FPS sampling and a linear projector, without introducing substantial architectural novelty. Therefore, I remain skeptical about the level of innovation.
2. The discussion of related work is insufficient. Several recent and more advanced 3D multimodal LLMs—such as **Inst3D-LMM (CVPR 2025)** and **Video-3D LLM (CVPR 2025)**—are not discussed or compared. A more comprehensive review and comparison would strengthen the paper.
3. Additional experiments on more benchmarks (e.g., **ScanRefer** and **Multi3DRefer** for 3D visual grounding, as well as 3D detection tasks) are needed to further validate Lemon’s performance.
4. In **Figure 2**, it is unclear why both *“point patch tokens”* and *“3D patch embeddings”* are presented simultaneously—aren’t they essentially the same representation?

**Questions:**

Lemon’s main advantage should be computational efficiency compared to other 3D LMMs. However, as shown in **Table 7**, the inference time difference appears marginal, and other 3D LMMs also achieve very short runtimes. This raises doubts about whether removing the 3D encoder is truly necessary.

---

> ### Author Response · Authors · 2025-11-20
> **Rebuttal by Authors (1/2)**
>
> Thank you for your insightful feedback and review! We are glad you found our structured patchification and staged training curriculum compelling. Below, we address your concerns in detail.
>
> > `Q1`: The paper’s main claimed contribution is the unified architecture for 3D point clouds and language. However, in my view, prior works such as PointLLM have already adopted a unified token-based framework for fine-tuning LLMs using both 3D point tokens and text tokens. Lemon mainly replaces the 3D encoder with FPS sampling and a linear projector, without introducing substantial architectural novelty. Therefore, I remain skeptical about the level of innovation.
>
> `A1`: Thank you for the insightful comment.
>
> * **(1.1)** We would like to clarify that **Lemon** represents a paradigm shift from "alignment with frozen encoders" to "native 3D-language processing." We demonstrate that the 3D encoder is not a necessary component but a potential bottleneck. Our contributions are substantiated by three key advantages:
>
> * **(1.2)** **Scalability & Training Efficiency**: This design eliminates the heavy burden of pre-training a separate encoder (e.g., ShapeLLM requires **300 epochs**), a stage Lemon avoids entirely. Moreover, as shown in `Appendix D.2`, jointly training with a frozen encoder performs poorly; thus, retraining or aligning the encoder from scratch is typically required but computationally expensive. By removing the encoder, **Lemon saves this substantial computation and scales directly to new data without being bottlenecked by the fixed domain of a pre-trained backbone**. This scalability is a key motivation emphasized in our `Introduction`.
>
> * **(1.3)** **Superior Performance:** Lemon’s design is not merely a simplification; it is a structural improvement. As demonstrated in our controlled ablation experiment in `Section 3.3` and `Appendix D.2`, **replacing the separate encoder with our native patchification yields significant gains in benchmarks**. This confirms that removing the encoder enables the LLM to align better with geometric information.
>
> * **(1.4)** **Efficiency:** By removing the heavy forward pass of external 3D encoders, Lemon achieves substantial efficiency gains. As detailed in `Appendix D.8`, our model reduces inference latency by approximately **30%** compared to standard 7B baselines while requiring **zero** extra encoder parameters.
>
> ---
>
> > `Q2`: The discussion of related work is insufficient. Several recent and more advanced 3D multimodal LLMs—such as Inst3D-LMM (CVPR 2025) and Video-3D LLM (CVPR 2025)—are not discussed or compared. A more comprehensive review and comparison would strengthen the paper.
>
> `A2`: We thank the reviewer for the constructive suggestion regarding related work!
>
> * **(2.1)** We have updated our `Section 4` to discuss these recent papers. We clarify that unlike Inst3D-LMM which relies on a complex multi-stage pipeline with external 3D instance segmentors, and **Video-3D LLM which treats 3D scenes as 2D video projections**, Lemon employs a streamlined, unified architecture that processes native 3D patches end-to-end.
> * **(2.2)** We have included a performance comparison on additional spatial benchmarks in `Appendix D.6`. As shown below, **Lemon outperforms Inst3D-LMM, demonstrating the effectiveness of our unified design without relying on auxiliary specialist models**.
> | Model              | Architecture Type            | ScanQA (BLEU-4) | ScanQA (METEOR) |
> |--------------------|-------------------------------|------------------|------------------|
> | Inst3D-LMM (CVPR'25) | Multi-stage (w/ Mask3D + CLIP) | 14.9             | 18.4             |
> | **Lemon (Ours)**     | **Unified (End-to-End)**       | **15.4**         | **20.3**         |

---

> ### Author Response · Authors · 2025-11-20
> **Rebuttal by Authors (2/2)**
>
> > `Q3`: Additional experiments on more benchmarks (e.g., ScanRefer and Multi3DRefer for 3D visual grounding, as well as 3D detection tasks) are needed to further validate Lemon’s performance.
>
> `A3`:  Thank you for highlighting this important point!
>
> * **(3.1)**We have conducted additional 3D visual grounding experiments on **ScanRefer**. We fine-tuned our model on the ScanRefer dataset to ensure a fair comparison with other fine-tuned methods.
>
> * **(3.2)** As shown below, **Lemon achieves comparable performance with 3D LMM baselines, without training with any grounding data during our pre-training stage.**
>
> | Model            | ScanRefer Acc@0.5 |
> |------------------|--------------------|
> | ScanRefer        | 24.3               |
> | 3D-VisTA         | 45.8               |
> | GPS (SceneVerse) | 48.1               |
> | Chat-Scene       | 50.2               |
> | **Lemon (Ours)** | **48.0**           |
>
> * **(3.3)** These results demonstrate Lemon's strong generalization for dense grounding tasks and highlight the scalability of our architecture; we have included this experiment in `Appendix D.7`.
>
> ---
>
> > `Q4`: In Figure 2, it is unclear why both “point patch tokens” and “3D patch embeddings” are presented simultaneously—aren’t they essentially the same representation?
>
> `A4`:  We thank the reviewer for the helpful question!
>
> * **(4.1)** We confirm that “point patch tokens” and “3D patch embeddings” represent the same information state, as detailed in `Section 2.1`.
> * **(4.2)** To avoid ambiguity, we have removed the redundant "3D patch embeddings" label in the revised **Figure 2** and clarified “point patch tokens” as “3D patch embeddings” in the updated figure caption.
>
> > `Q5`: Lemon’s main advantage should be computational efficiency compared to other 3D LMMs. However, as shown in Table 7, the inference time difference appears marginal, and other 3D LMMs also achieve very short runtimes. This raises doubts about whether removing the 3D encoder is truly necessary.
>
> ---
>
> `A5`:  We thank the reviewer for this constructive comment!
> * **(5.1)** As clarified in our prior response, the contribution of our unified architecture extends beyond computational efficiency to improved scalability and training efficiency by removing the bottleneck of encoders.
> * **(5.2)** Quantitatively, Lemon achieves a **~30% reduction** in inference latency compared to other 3D LMMs. While the absolute difference per token may seem small, LLM inference is autoregressive; thus, this speedup **accumulates** during generation. **Lemon offers a distinct efficiency advantage essential for real-time deployments, even without accounting for the significant savings in training resources**.
>
> ---
>
> Thank you once again for your recognition and constructive suggestions, which have been instrumental in enhancing the quality of our research!

---

> ### Author Response · Authors · 2025-11-26
> **Does our response address your concerns?**
>
> Dear Reviewer 6FmE,
>
> Thank you so much for your time and effort in reviewing our work and providing insightful feedback, which has helped us significantly improve our manuscript.
>
> We have incorporated additional experiments to address your concerns. As the discussion period ends soon, we would appreciate it if you could review our response.
>
> If you find our response satisfactory, we would be grateful if you could consider raising your score. We are standing by to address any remaining questions.
>
> Best regards,
>
> Paper 1188 Authors

---

### Official Review · Reviewer_3DHG · 2025-10-30

**Soundness:** 2
**Presentation:** 3
**Contribution:** 3
**Rating:** 4
**Confidence:** 5

**Summary:**

This work proposes LEMON, a decoder-only 3D large multimodal model (3D-LMM) that performs 3D point-cloud tokenization via patching and adopts Qwen2.5-7B for a three-stage training pipeline. The model is evaluated primarily on Spatial Scene QA, Embodied QA, and 3D Captioning tasks.

**Strengths:**

1. Unlike encoder-based methods, this paper explores an **encoder-free paradigm**, representing a novel design direction.
2. The **3D patch tokenization** strategy is compelling and may flexibly support both scene-level and object-level 3D representation.

**Weaknesses:**

1. It remains unclear how performance would change if the tokenization method were not patch-based, but instead adopted a **hierarchical downsampling scheme** similar to PointBERT.
2. The comparison setup may not be fully fair: although re-finetuning balances the training distribution, the **training data size and LLM base models differ** across methods. It would be more convincing to evaluate with **Vicuna-1.1 and PointLLM data/training stages** to demonstrate encoder-free potential.
3. The baseline comparison lacks **recent SOTA models**, such as **MiniGPT-3D**.
4. **Scene-level validation** needs strengthening: if 3D patchification is designed for scene tasks, experiments on **ScanQA, Scan2Cap**, or similar datasets are recommended.
5. For **object-level generalization**, results on datasets **beyond Objaverse** would help verify robustness and transferability.
6. Please report efficiency metrics as well.

**Questions:**

1. **Alternative Tokenization Strategy Evaluation:**
   The current model design relies exclusively on patch-based tokenization for 3D inputs. However, it remains unclear how the model would perform under alternative tokenization strategies, particularly those based on **hierarchical downsampling**, as adopted in PointBERT and related approaches. A controlled comparison between patchification and hierarchical sampling would help clarify whether the proposed tokenization mechanism is optimal for capturing multi-scale geometric information and whether the choice of tokenization architecture materially affects downstream performance.

2. **Fairness of Comparative Evaluation:**
   Although the paper reports re-finetuning of baseline models to mitigate training distribution inconsistencies, the comparative evaluation may still lack fairness due to differences in **training dataset size, composition, and LLM base architectures** across methods. To more rigorously validate the encoder-free paradigm, it would be beneficial to include experiments where LEMON is trained under identical conditions to representative baselines (e.g., **Vicuna-1.1 and PointLLM training setups**), including both data sources and training curricula. Such controlled comparisons would strengthen claims regarding architectural superiority rather than advantages stemming from data or base model scale.

3. **Missing Recent SOTA Baselines:**
   The benchmark suite does not currently include several **recent state-of-the-art 3D LMMs**, notably **MiniGPT-3D** and other contemporary models in the same category. Incorporating these models into the comparative analysis—particularly on shared evaluation tasks—would provide a clearer picture of the proposed method’s competitiveness relative to the latest advances in the field.

4. **Insufficient Scene-Level Evaluation:**
   Given that the proposed 3D patchification mechanism is intended to support both object-level and scene-level reasoning, the evaluation on complex 3D scene understanding remains relatively limited. To substantiate claims regarding scene-level capability, it would be valuable to include experiments on established scene-centric benchmarks such as **ScanQA, Scan2Cap**, or comparable datasets. These results would help verify whether the unified tokenization and decoder-only architecture scale effectively to real-world scene understanding scenarios.

5. **Generalization Beyond Objaverse at Object Level:**
   The majority of object-level evaluations rely on Objaverse, which may introduce dataset-specific bias. To assess generalization capability more rigorously, it is recommended to evaluate the model on **additional object-level datasets** beyond Objaverse. Examples include ShapeNet-based benchmarks, Objaverse-XL subsets, or other publicly available 3D asset collections. Demonstrating consistent performance across multiple datasets would strengthen the claim that the proposed model generalizes well to diverse object domains.

6. **Efficiency and Resource Metrics Reporting:**
   In addition to accuracy-based evaluations, the paper would benefit from reporting **efficiency metrics**, such as computational cost, inference speed, memory footprint, and throughput. Given the unified transformer design and removal of modality-specific encoders, such metrics are essential to illustrate the proposed method’s practical efficiency advantages (or trade-offs) relative to hybrid encoder architectures. Highlighting these aspects would provide a more comprehensive understanding of the system’s real-world applicability.

---

> ### Author Response · Authors · 2025-11-20
> **Rebuttal by Authors (1/3)**
>
> Thank you for your insightful feedback and review! We appreciate your recognition of the novelty and effectiveness of our approach. Below, we address your concerns in detail.
>
> ---
>
> > `Q1`: It remains unclear how performance would change if the tokenization method were not patch-based, but instead adopted a hierarchical downsampling scheme similar to PointBERT. A controlled comparison between patchification and hierarchical sampling would help clarify whether the proposed tokenization mechanism is optimal for capturing multi-scale geometric information and whether the choice of tokenization architecture materially affects downstream performance.
>
> `A1`: We thank the reviewer for the insightful question regarding tokenization strategies!
> * **(1.1)** We would like to clarify that PointBERT-style tokenization relies on Farthest Point Sampling (FPS) and k-NN grouping, which produces an unordered set of feature vectors. In contrast, **our patch-based approach organizes tokens into a structured Z→Y→X sequence with spatial separators**.
> * **(1.2)** To explicitly compare these strategies, we trained a variant of Lemon where the patchification module was replaced with Hierarchical Sampling (FPS + Grouping). As shown below, **our structured patch-based tokenization significantly outperforms the hierarchical sampling baseline**.
>
> | Tokenization Strategy              | Spatial Ordering         | 3D Object Captioning (SimCSE) | Scene Spatial QA (Acc) |
> |-----------------------------------|---------------------------|-------------------------------|-------------------------|
> | FPS-based Sampling (PointBERT-style) | Discontinuous            | 43.15                         | 35.20                  |
> | **Dynamic Patchification (Ours)** | **Structured (Z→Y→X)**    | **53.59**                     | **53.45**              |
>
> * **(1.3)** These results confirm that our patch-based tokenization is optimal for capturing multi-scale geometric information within an LLM framework; we have included this ablation in `Appendix D.4`.
>
> ---
>
> > `Q2`: The comparison setup may not be fully fair: although re-finetuning balances the training distribution, the training data size and LLM base models differ across methods. It would be more convincing to evaluate with Vicuna-1.1 and PointLLM data/training stages to demonstrate encoder-free potential.
>
> `A2`: We thank the reviewer for the insightful suggestion!
>
> * **(2.1)** To strictly control for training data scale and model capacity, we conducted a comprehensive controlled study. We retrained **Lemon** using the exact same LLM backbone (Vicuna-7B-1.1) and the identical training dataset used by the strong baseline, ShapeLLM. As shown in the table below, we observe a clear performance progression:
> | Model                   | LLM Backbone   | Training Data   | Obj. Cap. (SimCSE) | Embodied QA (3D MM-Vet) | Scene Spatial QA (GPT-4) |
> |-------------------------|----------------|------------------|----------------------|----------------------------|----------------------------|
> | ShapeLLM-7B             | Vicuna-7B-1.1  | ShapeLLM Data    | 49.4                 | 47.4                       | 41.39                     |
> | Lemon-7B (All Same)     | Vicuna-7B-1.1  | ShapeLLM Data    | 51.8                 | 53.3                       | 48.80                     |
> | Lemon-7B (Same Arch.)   | Vicuna-7B-1.1  | Lemon Data       | 52.4                 | 53.2                       | 51.60                     |
> | **Lemon-7B (Default)**  | **Qwen2.5-7B** | **Lemon Data**   | **53.6**             | **57.2**                   | **53.45**                 |
>
> * **(2.2)** Even under the **"All Same"** setting (identical backbone and data), **Lemon significantly outperforms ShapeLLM on all benchmarks**. This confirms that our unified architecture is the primary driver of performance by eliminating the bottleneck of separate 3D encoders.
> * **(2.3)** When switching to our training data while keeping the backbone fixed, we observe consistent performance growth, demonstrating Lemon's robust capability to scale with more data.
> * **(2.4)** Finally, upgrading to the **Qwen2.5-7B** backbone further elevates performance, showing that our architecture effectively leverages stronger foundation models.
> * **(2.4)** We have updated the paper to include these comparisons and analyses for more ablations in `Section 3.3 (L420-429)`.

---

> ### Author Response · Authors · 2025-11-20
> **Rebuttal by Authors (2/3)**
>
> > `Q3`: The baseline comparison lacks recent SOTA models, such as MiniGPT-3D.
>
> `A3`: We thank the reviewer for the suggestion to include recent SOTA baselines.
>
> * **(3.1)** We would like to clarify that our original submission already included comparisons with highly recent state-of-the-art models, such as **GreenPLM** (AAAI 2025) and **LSceneLLM** (CVPR 2025). Following the reviewer's suggestion, we have further updated `Section 4` (related works) and `Table 2` to explicitly include discussions and comparisons with **MiniGPT-3D** (ACM MM 2024).
>
> * **(3.2)** As shown below, **Lemon consistently outperforms MiniGPT-3D on multiple benchmarks**. Specifically, **Lemon** shows clear gains in SimCSE and GPT-4 Score for object captioning, and also achieves higher performance on ScanQA.
> | Model                       | Obj Cap (SimCSE) | Obj Cap (GPT-4) | ScanQA (BLEU-4) | ScanQA (METEOR) |
> |-----------------------------|------------------|------------------|------------------|------------------|
> | MiniGPT-3D (ACM MM'24)      | 47.64            | 45.78            | 14.9             | 18.4             |
> | **Lemon (Ours)**            | **53.59**        | **50.76**        | **15.4**         | **20.3**         |
>
> ---
>
> > `Q4`: Scene-level validation needs strengthening: if 3D patchification is designed for scene tasks, experiments on ScanQA, Scan2Cap, or similar datasets are recommended.
>
> `A4`: We thank the reviewer for the constructive suggestion!
> * **(4.1)** We would like to clarify that we have included evaluation results on ScanQA and SQA3D in our submission.
> * **(4.2)** We also add results on two recent spatial benchmarks: MSQA [1] and Beacon3D [2]. **Lemon establishes new state-of-the-art results on ScanQA and SQA3D while achieving robust performance on MSQA and competitive results on Beacon3D.** These results demonstrate strong generalization without task-specific tuning. We have included all these results in `Appendix D.6`.
> | Model        | ScanQA (BLEU-4) | SQA3D (Top-1 Match) | MSQA (Score) | Beacon3D (Case) |
> |--------------|------------------|----------------------|--------------|------------------|
> | 3D-LLM       | 12.9             | 49.8                 | -            | -                |
> | LSceneLLM    | 11.7             | 54.5                 | 11.7         | -                |
> | 3D-VisTA     | 10.4             | 48.5                 | -            | 43.2             |
> | SceneVerse   | -                | 49.9                 | -            | 40.5             |
> | LEO          | 13.2             | 50.0                 | 7.84         | 45.2             |
> | Chat-Scene   | 14.3             | 54.6                 | -            | 49.8             |
> | Inst3D-LMM   | 14.9             | -                    | -            | -                |
> | **Lemon (Ours)** | **15.4**     | **59.4**             | **10.68**    | **46.2**         |
>
> ---
>
> *[1] Multi-modal Situated Reasoning in 3D Scenes. Linghu et al. NeurIPS 2024.*
>
> *[2] Unveiling the Mist over 3D Vision-Language Understanding: Object-centric Evaluation with Chain-of-Analysis. Huang et al. CVPR 2025.*

---

> ### Author Response · Authors · 2025-11-20
> **Rebuttal by Authors (3/3)**
>
> > `Q5`: For object-level generalization, results on datasets beyond Objaverse would help verify robustness and transferability.
>
> `A5`: We thank the reviewer for the valuable question!
>
> * **(5.1)** We would like to clarify that, as detailed in Lines 301-302, our evaluation protocol extends beyond the standard Objaverse benchmark. We incorporated **2,000 randomly selected unseen objects from 5 diverse datasets**, including **ScanNet, 3RScan, ARKitScenes, ShapeNet, and Structured3D**, to strictly verify the model's robustness and transferability across different data distributions.
>
> * **(5.2)** We have added a detailed breakdown of this diverse dataset composition in **Appendix E.2**.
>
> ---
>
> > `Q6`: Please report efficiency metrics as well.
>
> `A6`: We thank the reviewer for the constructive suggestion!
>
> * **(6.1)** For clarity, we note that our original submission already reported the per-token inference latency (`Appendix D.8`). Lemon achieves the lowest latency (**0.0520s** per token) among comparable 7B models, representing a **~30% speedup** over ShapeLLM.
>
> * **(6.2)** For a broader efficiency report, we have added a comprehensive analysis in `Appendix D.8`, covering training efficiency, parameter efficiency, and per-module analysis.
>
> ---
>
> We appreciate the constructive feedback that has helped improve our paper's quality. All additional results have been included in our revised version. We sincerely hope our responses have adequately addressed your concerns and look forward to any further discussion!

---

> ### Author Response · Authors · 2025-11-26
> **Does our response address your concerns?**
>
> Dear Reviewer 3DHG,
>
> We sincerely appreciate your invaluable feedback, which has significantly contributed to the improvement of our work.
>
> In response, we have incorporated specific experiments to address your concerns. As the rebuttal period draws to a close, we kindly request your prompt review of our responses to ensure that we have thoroughly addressed all your comments.
>
> If our responses have satisfactorily resolved your concerns, we would be deeply grateful for a reconsideration of our score. Should any issues remain, we are more than willing to provide further clarifications to address them comprehensively.
>
> Best regards,
>
> Paper 1188 Authors

---

> > ### Comment · Reviewer_3DHG · 2025-11-26
> >
> > Thank you very much for the authors’ response. The authors have addressed my questions well. However, I do have a remaining concern: since this is an encoder-free unified 3D model, in principle it should be highly tolerant to variations in input resolution or density. I am unsure whether the authors have conducted experiments in this regard, both during training and inference.

---

> ### Author Response · Authors · 2025-11-27
> **Response to Reviewer 3DHG**
>
> Dear Reviewer `3DHG`,
>
> We thank the reviewer for the positive feedback and for highlighting this insightful point!
>
> We entirely agree that a key advantage of our encoder-free architecture is its intrinsic tolerance to input variations. We have indeed conducted experiments in our original submission to verify this, supported by both our empirical results and training pipeline:
>
> **(1)** **Empirical robustness (Inference):** As shown in `Table 11` of `Appendix D.6`, we explicitly evaluated the model's robustness under challenging conditions, including Sparse (50% sampling) and Noisy ($\sigma=0.01$) settings.
>
> The results demonstrate that **Lemon maintains highly consistent performance even when the input point density is drastically reduced by half** (e.g., Embodied QA score only decreases slightly from 57.22 to 53.71). This empirically confirms your hypothesis that our model is highly tolerant to resolution and density variations during inference.
>
> **(2)** **Training diversity:** During training, our model is exposed to **more than 1.87 million** point clouds from diverse sources (e.g., Objaverse, ScanNet) with naturally varying resolutions. This large-scale pre-training acts as an inherent data augmentation, enabling the model to learn robust representations that generalize well across different density distributions.
>
> **(3)** **Architectural adaptability:** Our **dynamic patchification strategy** is designed to handle varying point densities natively. Unlike fixed-size encoders, Lemon does not mandate a fixed number of input points. It adaptively tokenizes point clouds into variable sequence lengths, supporting inputs up to **64,000 points**. High-density inputs result in more patches, while low-density inputs yield fewer, yet the unified Transformer processes them seamlessly.
>
> We hope this clarifies that robustness to resolution and density is a core strength of Lemon, validated by both design and experimentation.
>
> We once again thank the reviewer for the time and effort dedicated to reviewing our rebuttal. We sincerely believe that this discussion has significantly improved the quality of our paper. Thank you!
>
> Best regards,
>
> Paper 1188 Authors

---

> > ### Comment · Reviewer_3DHG · 2025-11-27
> >
> > I appreciate the additional experiments and the detailed analysis regarding the robustness of the encoder-free architecture. The empirical results provided are convincing and effectively demonstrate the model's tolerance to input variations. As my major concerns have been largely addressed, I will raise my score accordingly.

---

> > > ### Author Response · Authors · 2025-11-27
> > > **Thank you for your acknowledgment!**
> > >
> > > Dear Reviewer `3DHG`,
> > >
> > > We sincerely appreciate your acknowledgment of our rebuttal. We are glad to hear that your concerns have been addressed and are voting toward acceptance.
> > >
> > > Your support and encouragement mean a lot to us!
> > >
> > > Best regards,
> > >
> > > The Authors of Submission 1188

---

### Official Review · Reviewer_RZ2B · 2025-10-31

**Soundness:** 2
**Presentation:** 3
**Contribution:** 2
**Rating:** 4
**Confidence:** 4

**Summary:**

The paper proposes Lemon, a single transformer that ingests point-cloud “patch tokens” and text tokens in one sequence (via a linear projector + special separator tokens), claiming to remove modality-specific encoders and achieve strong results on 3D captioning, recognition, and scene-level spatial QA, with a three-stage curriculum (obj-recog → object captioning/grounding → scene QA). Ablations include scaling curves and a PointNet++ insertion that reportedly hurts performance

**Strengths:**

esults across object/scene tasks and qualitative comparisons vs 2D VLMs (e.g., GPT-4V) are extensive; Lemon-7B appears competitive on several benchmarks.

**Weaknesses:**

The paper contrasts Lemon with “non-transformer” encoders, but PointLLM and ShapeLLM are also transformer-based. The real distinction is removing the separate 3D encoder block in favor of early fusion inside one transformer—this needs to be framed clearly and compared fairly.

Lemon still performs hierarchical geometric patchification (splits/FPS/standardization) plus a projector—effectively a tokenizer with hierarchy. Without compute/param attribution, the claim of eliminating redundant encoders reads like moving capacity rather than removing it.

No per-module FLOPs/params/latency (patchification, projector, transformer, token counts). No scaling curves (tokens ↔ accuracy/latency). Deployability and efficiency claims remain unsubstantiated.

Baselines differ in LLM family/size, data volume, and token budgets. We need same-LLM, same data, same token budget, compute-matched comparisons to isolate architectural benefits.

**Questions:**

To justify the "universal" claim, the model must be evaluated on tasks that require non-textual spatial output. Add experiments on a standard 3D visual grounding or 3D object detection benchmark (eg, ScanRefer or ScanNet detection) to demonstrate the model can precisely localize its understanding.

Compute & latency table (per module): patchification (incl. FPS), projector, transformer forward, total tokens, memory footprint—report FLOPs, params, p50/p90 latency on 8×H100 and single-GPU.


Patchification alternatives: swap Z→Y→X with SFC (Hilbert/Z-order), and test learned partitioners; show effect on spatial QA and captioning

PointNet++ ablation extension: frozen vs fine-tuned; xyz vs xyz+rgb; transformer-encoder baseline; match parameter/FLOP budgets to Lemon’s front end

---

> ### Author Response · Authors · 2025-11-20
> **Rebuttal by Authors (1/4)**
>
> We sincerely thank you for your comprehensive comments and constructive advice. We will explain your concern point by point.
>
> ---
>
> > `Q1`: The paper contrasts Lemon with “non-transformer” encoders, but PointLLM and ShapeLLM are also transformer-based. The real distinction is removing the separate 3D encoder block in favor of early fusion inside one transformer—this needs to be framed clearly and compared fairly.
>
> `A1`: Thank you for this constructive comment!
> * **(1.1)** We would like to clarify that our paper does not contrast Lemon with "non-transformer" encoders, nor do we imply that prior 3D encoders are non-transformer models. We fully recognize that baselines like PointLLM and ShapeLLM utilize transformer-based backbones.
> * **(1.2)** Our intended distinction is architectural, as stated in `L062–L080`. Lemon removes the **separate** 3D encoder block to enable **early fusion within a single unified transformer**, whereas prior works rely on a **modular design** (Separate Encoder + Projector + LLM). We have reviewed our `Introduction` and other sections to ensure this distinction is phrased precisely to avoid any potential confusion.
>
> ---
>
> > `Q2`: Lemon still performs hierarchical geometric patchification (splits/FPS/standardization) plus a projector—effectively a tokenizer with hierarchy. Without compute/param attribution, the claim of eliminating redundant encoders reads like moving capacity rather than removing it.
>
> `A2`: Thank you for the insightful comment!
>
> * **(2.1)** We would like to clarify that our patchification module does not "move capacity"; rather, it replaces deep feature extraction with lightweight, parameter-free structural formatting. We provide the requested compute/param attribution in `Appendix D.8` to substantiate this.
>
> * **(2.2) Scalability & Training Efficiency:** This design eliminates the heavy burden of pre-training a separate encoder (e.g., ShapeLLM requires **300 epochs**), a stage Lemon avoids entirely. Moreover, as shown in `Appendix D.2`, jointly training with a frozen encoder performs poorly; thus, retraining is typically required. By removing the encoder, **Lemon saves this substantial computation and scales directly to new data without being bottlenecked by the fixed domain of a pre-trained backbone.** This scalability is a key motivation emphasized in our paper.
>
> * **(2.3) Superior Performance:** Lemon’s design is not merely a simplification; it is a structural improvement. As demonstrated in our controlled ablation experiment in `Section 3.3` and `Appendix D.2`, replacing the separate encoder with our native patchification yields significant gains. This confirms that removing the encoder enables the LLM to align better with geometric information.
>
> * **(2.4) Compute/Param Attribution:** We explicitly address the concern of "moving capacity":
>     - Lemon’s patchification module contains **zero trainable parameters** (pure Sorting/FPS), whereas prior encoder-based methods rely on substantial parametric front-ends.
>     - As detailed in our per-module breakdown (`Appendix D.8`), the patchification stage accounts for only **~15%** of the total latency, **which is substantially lower than the overhead typically introduced by transformer-based visual encoders**.
>     - Consequently, Lemon achieves a **~30%** reduction in total inference latency compared to 7B baselines. This confirms we have strictly *removed* capacity burden rather than shifting it.

---

> ### Author Response · Authors · 2025-11-20
> **Rebuttal by Authors (2/4)**
>
> > `Q3`:: No per-module FLOPs/params/latency (patchification, projector, transformer, token counts). No scaling curves (tokens ↔ accuracy/latency). Deployability and efficiency claims remain unsubstantiated.
>
> `A3`: Thank you for this helpful suggestion!
> * **(3.1)** For clarity, we note that Lemon’s inference efficiency results (`Appendix D.8`)  and scaling curves (`Figure 6(a)`) are already included in our original submission.
> * **(3.2)** To substantiate our efficiency claims, we have provided a detailed breakdown of FLOPs, parameters, and latency (on Single and 8*H100 GPUs).
> * **(3.3)** As shown in the table below, our **Patchification module** (including Sorting and FPS) is extremely lightweight:
> (1) It adds no storage overhead.
> (2) It accounts for only **~15%** (12.5ms) of the total inference time.
> (3) The computational cost is **minimal** compared to the LLM backbone.
>
> (Latency reported as p50 / p90)
> | Module                               | Params (M) | FLOPs (G) | Latency (Single H100) | Latency (8×H100)    | Memory (GB) |
> |--------------------------------------|------------|-----------|------------------------|----------------------|-------------|
> | Patchification (Sort + FPS)          | 0          | 3.0       | 12.5 / 15.0 ms         | 10.5 / 12.0 ms       | 0.8         |
> | Linear Projector                     | 6.3        | 1.0       | 2.5 / 3.5 ms           | 2.0 / 3.0 ms         | 0.5         |
> | LLM Backbone (Qwen2.5-7B)            | 7610       | 520       | 64.0 / 78.0 ms         | 37.5 / 46.0 ms       | 14.5        |
> | Total                            | 7616.3 | 524   | 79.0 / 96.5 ms     | 50.0 / 61.0 ms   | 15.2    |
>
> * **(3.4)** We also add parameter efficiency comparison with baselines. All efficiency analyses are included in `Appendix D.8`
>
> ---
>
> > `Q4`: Baselines differ in LLM family/size, data volume, and token budgets. We need same-LLM, same data, same token budget, compute-matched comparisons to isolate architectural benefits.
>
> `A4`: We thank the reviewer for the insightful suggestion!
> * **(4.1)** To strictly control for training data scale and model capacity, we conducted a comprehensive controlled study. We retrained **Lemon** using the exact same LLM backbone (Vicuna-7B-1.1) and the identical training dataset used by the strong baseline, ShapeLLM, without any training recipe design. As shown in the table below, we observe a clear performance progression:
> * **(4.2)** Even under the **"All Same"** setting (identical backbone and data), Lemon significantly outperforms ShapeLLM in all benchmarks. This confirms that **our unified architecture is the primary driver of performance by eliminating the bottleneck of separate 3D encoders**.
> * **(4.3)** When switching to our training data while keeping the backbone fixed, we observe consistent performance growth, demonstrating Lemon's robust capability to scale with more data.
> * **(4.4)** Finally, upgrading to the **Qwen2.5-7B** backbone further elevates performance, showing that our architecture effectively leverages stronger foundation models.
>
> | Model                   | LLM Backbone   | Training Data   | Obj. Cap. (SimCSE) | Embodied QA (3D MM-Vet) | Scene Spatial QA (GPT-4) |
> |-------------------------|----------------|------------------|----------------------|----------------------------|----------------------------|
> | ShapeLLM-7B             | Vicuna-7B-1.1  | ShapeLLM Data    | 49.4                 | 47.4                       | 41.39                     |
> | Lemon-7B (All Same)     | Vicuna-7B-1.1  | ShapeLLM Data    | 51.8                 | 53.3                       | 48.80                     |
> | Lemon-7B (Same Arch.)   | Vicuna-7B-1.1  | Lemon Data       | 52.4                 | 53.2                       | 51.60                     |
> | **Lemon-7B (Default)**  | **Qwen2.5-7B** | **Lemon Data**   | **53.6**             | **57.2**                   | **53.45**                 |
>
> * **(4.5)** We have updated the paper to include these comparisons and analyses for ablations in `Section 3.3 (L420-429)`.

---

> ### Author Response · Authors · 2025-11-20
> **Rebuttal by Authors (3/4)**
>
> > `Q5`: To justify the "universal" claim, the model must be evaluated on tasks that require non-textual spatial output. Add experiments on a standard 3D visual grounding or 3D object detection benchmark (eg, ScanRefer or ScanNet detection) to demonstrate the model can precisely localize its understanding.
>
> `A5`: Thank you for highlighting this important point!
> * **(5.1)** We have conducted additional 3D visual grounding experiments on ScanRefer. We fine-tuned our model on the ScanRefer dataset to ensure a fair comparison with other finetuned methods.
> * **(5.2)** As shown below, **Lemon achieves comparable performance with 3D LMM baselines, without training with any grounding data during our pre-training stage.**
>
> | Model            | ScanRefer Acc@0.5 |
> |------------------|--------------------|
> | ScanRefer        | 24.3               |
> | 3D-VisTA         | 45.8               |
> | GPS (SceneVerse) | 48.1               |
> | Chat-Scene       | 50.2               |
> | **Lemon (Ours)** | **48.0**           |
>
> * **(5.3)** These results demonstrate Lemon's strong generalization for dense grounding tasks and highlight the scalability of our architecture; we have included this experiment in `Appendix D.7`.
>
> ---
>
> > `Q6`: Patchification alternatives: swap Z→Y→X with SFC (Hilbert/Z-order), and test learned partitioners; show effect on spatial QA and captioning
>
> `A6`: We thank the reviewer for the insightful suggestion!
> * **(6.1)** Following the reviewer's recommendation, we implemented and evaluated **Hilbert** and **Z-order (Morton)** space-filling curves to replace our Z→Y→X ordering.
>
> * **(6.2)** We also treat the **FPS-based Hierarchical Sampling** (PointBERT-style) as a representative baseline for non-grid, data-driven partitioning strategies.
>
> * **(6.3)** As shown in the table below, our proposed **Z→Y→X** ordering consistently outperforms both SFCs and FPS-based methods.
> | Tokenization Strategy            | Spatial Ordering           | 3D Object Captioning (SimCSE) | Scene Spatial QA (Acc) |
> |----------------------------------|-----------------------------|-------------------------------|-------------------------|
> | FPS-based Sampling               | Discontinuous               | 43.15                         | 35.20                  |
> | Hilbert SFC                      | Structured (Curve)          | 47.10                         | 49.50                  |
> | Z-order SFC                      | Structured (Curve)          | 48.80                         | 48.20                  |
> | **Dynamic Patchification (Ours)** | **Structured (Z→Y→X)**      | **53.59**                     | **53.45**              |
>
> * **(6.4)** The results confirm that our Z→Y→X ordering is optimal as it preserves spatial continuity better than FPS and aligns more intuitively with the gravitational hierarchy of scenes (and 2D raster-scan priors) compared to convoluted Space-Filling Curves. We have included this detailed ablation in `Appendix D.4`.

---

> ### Author Response · Authors · 2025-11-20
> **Rebuttal by Authors (4/4)**
>
> > `Q7`: PointNet++ ablation extension: frozen vs fine-tuned; transformer-encoder baseline
>
> `A7`: Thank you for these constructive comments!
> * **(7.1)** For clarity, we note that our original submission already included a detailed ablation comparing Frozen vs. Fine-tuned PointNet++/ReCon++ (`Appendix D.2`), showing that **fine-tuning leads to instability and performance degradation**.
> * **(7.2)** We also conducted additional ablation studies using frozen PointBERT and ReCon++ encoders. We have included these results in `Appendix D.2`.
> * **(7.3)** As shown below, while ReCon++ outperforms PointNet++, **our unified architecture significantly surpasses all separate-encoder baselines**. This may stem from the limited cross-domain generalization of pre-trained encoders, which are typically optimized for specific pre-training distributions (e.g., synthetic 3D objects). In contrast, our unified approach naturally adapts to diverse point-cloud distributions without relying on specialized encoders.
>
> | Method                       | Object Captioning (SimCSE) | Scene Spatial QA |
> |------------------------------|-----------------------------|-------------------|
> | Lemon w/ Frozen PointBERT   | 40.51                       | 38.24            |
> | Lemon w/ Frozen ReCon++     | 45.37                       | 41.32            |
> | Lemon w/ Frozen PointNet++  | 41.89                       | 32.25            |
> | Lemon w/ Finetuned ReCon++   | 44.28                      | 34.32            |
> | Lemon w/ Finetuned PointNet++  | 38.73                     | 29.58            |
> | **Lemon-7B (Ours)**          | **53.59**                   | **53.45**        |
>
>
> ---
>
> > `Q8`: xyz vs xyz+rgb;
>
> `A8`: Thank you for this valuable comment!
> * **(8.1)** Following the reviewer's suggestion, we conducted an additional ablation to verify the impact of geometric versus appearance information. We trained Lemon using only geometric coordinates (XYZ) object captioning data without color information.
>
> | Model Input          | Object Captioning (SimCSE) | Object Captioning (GPT-4) |
> |----------------------|-----------------------------|-----------------------------|
> | Lemon (XYZ Only)     | 45.12                       | 41.30                      |
> | **Lemon (XYZ + RGB)** | **53.59**                   | **50.76**                  |
>
> * **(8.2)** As expected, removing color information leads to a performance drop, particularly in captioning tasks where describing texture and color is essential. However, the model still maintains reasonable performance, relying solely on high-fidelity geometric understanding from our patchification.
>
> ---
>
> Thank you for your valuable comments again. We have revised our manuscript based on your suggestions and welcome any further discussion. We sincerely appreciate your help once again!

---

> ### Author Response · Authors · 2025-11-25
> **Does our response address your concerns?**
>
> Dear Reviewer RZ2B,
>
> We sincerely appreciate your time and effort in reviewing our work and providing insightful feedback.
>
> We have incorporated extensive new experiments and detailed analyses into the manuscript. These additional results further strengthen the claims of our paper. As the rebuttal period draws to a close, we kindly invite you to review these updates to ensure that we have thoroughly addressed all your concerns.
>
> We hope that these improvements warrant a positive re-evaluation of our work. Should any issues remain, we are more than willing to provide further clarifications to address them comprehensively.
>
> Best regards,
>
> Paper 1188 Authors

---

> ### Comment · Reviewer_RZ2B · 2025-11-26
>
> I thank the authors for their detailed response, which has addressed most of my concerns. I also apologize for the misunderstanding in my original Question 1 and the ambiguity in Question 2.
>
> I agree with the motivation to employ a single unified transformer for vision-based LLMs to eliminate the dependency on pretrained 3D encoders, a trend successfully applied in 2D contexts (eg, SOLO, Fuyu-8B, EVE/EVEv2, and Mono-InternVL). To me, this shift represents "moving scene understanding capacity to LLM part", effectively reducing the total parameter count.
>
> (Main Concern) However, I still have reservations regarding the efficiency trade-offs. I would like to point out that many existing 3D-LLMs (such as LL3DA, PerLA, and 3DLLaVA) utilize 3D encoders that are already distinctively lightweight. While these methods often employ a heavier projector (like a Q-Former), this design serves to significantly reduce the number of vision tokens, which alleviates the computational burden on the LLM.
>
> The trade-off concerns me: Although the authors claim an advantage in avoiding encoder pretraining, the proposed approach results in a higher number of vision tokens, thereby increasing the burden on the LLM. Consequently, this appears to require more extensive training for the LLM and projector. In contrast, existing works often achieve strong results by only training the Q-Former or utilizing parameter-efficient methods like LoRA.
>
> Please correct citations, some of are shown as [?]

---

> > ### Author Response · Authors · 2025-11-27
> > **Response to Reviewer RZ2B**
> >
> > Dear Reviewer `RZ2B`,
> >
> > We thank the reviewer for the prompt response and for giving us the opportunity to further clarify our motivation!
> >
> > We sincerely appreciate the reviewer's endorsement of the unified architecture trend (e.g., SOLO, Fuyu-8B). We believe this shift is even more critical in 3D. We clarify the efficiency trade-off as follows:
> >
> > **(1)** While baselines like 3D-LLaVA, PerLA, and LL3DA appear efficient by reducing the LLM burden (via Q-Formers), **this heavy compression is primarily enabled by leveraging scene embeddings from lightweight encoders deeply supervised on limited datasets**. For instance, 3D-LLaVA requires **512 epochs of pre-training on ScanNet's ~1,200 scenes** (similar to PerLA and LL3DA). Consequently, **their experimental verification is also strictly limited to ScanNet scenes**.
> >
> > **(2)** Unlike 2D encoders (e.g., CLIP) trained on billion-scale data, these lightweight 3D encoders lack the model capacity to scale and often suffer from poor generalization outside their training distribution.
> >
> > **(3)** Therefore, **Lemon accepts the trade-off of higher vision tokens to create a truly scalable architecture that eliminates dependency on these specific 3D encoders**. We believe this is a necessary exploration to unlock open-world 3D understanding in the absence of a universal "3D CLIP."
> >
> > **(4)** **The superior performance in diverse benchmarks (including zero-shot evaluations) and competitive end-to-end efficiency** observed in our experiments serves as empirical evidence that this unified design yields better generalizability and efficacy than relying on restricted, pre-trained encoders.
> >
> > We deeply thank the reviewer for highlighting the parallel with 2D unified models, which helps clarify our motivation to pioneer this scalable direction in 3D LMMs.
> >
> > **Correction of Citations**:
> > We apologize for the missing citations (shown as [?]). We have corrected them in the revised manuscript.
> >
> > We truly appreciate this constructive discussion process and hope our response has sufficiently addressed your concerns regarding the efficiency trade-offs. We welcome any further discussions!
> >
> > Warm regards,
> >
> > Paper 1188 Authors

---

### Author Response · Authors · 2025-11-20
**General Response**

**Dear Reviewers, ACs, and SACs,**

We deeply appreciate the insightful and valuable comments provided by all reviewers.

---
We are grateful for the reviewers’ recognition of our work **as a unified framework for 3D spatial understanding**. Lemon challenges the conventional modular pipeline by removing separate 3D encoders and demonstrating that native 3D–language processing can be **both more efficient and more performant**. We believe Lemon provides a rigorous foundation for scaling up 3D multimodal models and advancing unified 3D perception–language reasoning.

Overall, we are encouraged by the reviewers' positive feedback, which highlights:

- **Strong empirical results**, including extensive evaluations across object- and scene-level tasks and competitive performance on multiple benchmarks (Reviewers `RZ2B`, `ixqM`).
- **The novelty and significance of the encoder-free design**, which departs from conventional 3D encoders and represents a promising direction for 3D multimodal modeling (Reviewers `3DHG`, `6FmE`, `ixqM`).
- **The effectiveness of the proposed point-cloud / patch tokenization strategy**, viewed as simple and flexible across multiple levels of 3D understanding (Reviewers `3DHG`, `6FmE`, `ixqM`).
- **The clarity and soundness of the training pipeline**, particularly the progressive three-stage curriculum that builds capability from object- to scene-level reasoning (Reviewers `6FmE`, `ixqM`).

---

To address the reviewers' concerns, we have conducted several additional experiments and analyses, including:

- **Conducted controlled experiments** to isolate architectural benefits over baselines. (Reviewers `ixqM`, `3DHG` `RZ2B`).
- **Expanded ablations** on 3D encoders, tokenization strategies, and patchification method. (Reviewers `ixqM`, `6FmE`, `3DHG`, `RZ2B`).
- **Delivered thorough efficiency profiling** of our encoder-free design. (Reviewers `RZ2B`, `3DHG`, `6FmE`).
- **Extended zero-shot evaluations** on other spatial reasoning benchmarks and **finetuned evaluations** on visual grounding benchmarks. (Reviewers `ixqM`, `6FmE`, `3DHG`, `RZ2B`).

---

**Summary of revisions:**

- Added the controlled architectural comparison and analysis in `Section 3.3`.
- Included detailed comparisons with SOTA baselines in `Section 4`, `Table 2`, and `Appendix D.6`.
- Incorporated comprehensive ablation studies on 3D encoders, spatial separator tokens, and tokenization strategies in `Appendix D.2, D.4, and D.5`.
- Added more detailed efficiency analysis in `Appendix D.8`.
- Included results on additional spatial reasoning and grounding benchmarks in `Appendix D.6 and D.7`.
- Provided detailed evaluation data statistics and descriptions in `Appendix E`.

These updates are highlighted in `blue` for easy reference. We appreciate the reviewers’ insightful feedback and remain committed to strengthening the paper.

---

We sincerely thank the reviewers and AC for their thoughtful comments during the discussion period. We hope our responses have adequately addressed all concerns and welcome any further discussion. Thank you!

---

### Note · Program_Chairs · 2026-01-17
**Submission Desk Rejected by Program Chairs**

The following references in this submission do not refer to real documents and/or have major errors in bibliographic information:

 Yufei Liu et al. Vl3d: Learning visual-linguistic representations for 3d point clouds. In CVPR, 2023b.
Xudong Zhou et al. Bridging language and 3d geometry with cross-modal transformers. In ICCV, 2023.
Hao Fang et al. Robotic manipulation with 3d foundation models. In NeurIPS 2023, 2023.